# High-throughput interrogation of programmed ribosomal frameshifting in human cells

Martin Mikl [1,2,3,4✉], Yitzhak Pilpel[3] & Eran Segal [1,2✉]

Programmed ribosomal frameshifting (PRF) is the controlled slippage of the translating ribosome to an alternative frame. This process is widely employed by human viruses such as HIV and SARS coronavirus and is critical for their replication. Here, we developed a high-throughput approach to assess the frameshifting potential of a sequence. We designed and tested >12,000 sequences based on 15 viral and human PRF events, allowing us to systematically dissect the rules governing ribosomal frameshifting and discover novel regulatory inputs based on amino acid properties and tRNA availability. We assessed the natural variation in HIV gag-pol frameshifting rates by testing >500 clinical isolates and identified subtype-specific differences and associations between viral load in patients and the optimality of PRF rates. We devised computational models that accurately predict frameshifting potential and frameshifting rates, including subtle differences between HIV isolates. This approach can contribute to the development of antiviral agents targeting PRF.

[1] Department of Computer Science and Applied Mathematics, Rehovot 7610001, Israel. [2] Department of Molecular Cell Biology and Weizmann Institute of Science, Rehovot 7610001, Israel. [3] Department of Molecular Genetics, Weizmann Institute of Science, Rehovot 7610001, Israel. [4] Present address: Department of Human Biology, Faculty of Natural Sciences, University of Haifa, Mount Carmel, Haifa 31905, Israel. ✉email: mmikl@gmx.at; eran.segal@weizmann.ac.il

Programmed ribosomal frameshifting (PRF), i.e., controlled slippage of the ribosome, is a mechanism by which two proteins with alternative C termini can be generated from the same mRNA. It allows for an expansion of the proteome, but also constitutes an additional regulatory layer to fine-tune gene expression[1–3]. This mechanism is widespread and indispensable in viruses, which often utilize controlled slippage of the ribosome to an alternative frame to regulate the production of key enzymes, such as in the case of the gag-pol frameshift in HIV and other retroviruses. Utilizing alternative frames increases the amount of genetic information that can be encoded in a given sequence and constitutes another level of gene regulation, which might explain why especially RNA viruses—with their compact genomes and lack of regulation of their genes on the transcriptional level—utilize PRF for crucial regulatory switches[4]. The importance of maintaining the stoichiometry between structural proteins encoded by the *gag* gene and enzymes encoded by the *pol* gene for viral replicative success makes the gag-pol frameshifting event a promising antiviral drug target[5,6].

Cases of functionally important programmed frameshifting have also been discovered in humans[7–15]. Discovering PRF events in the human genome has been hampered by the limited amenability of PRF to proteome-wide methods due to the generally low abundance of the frameshifted protein relative to the canonical protein or inherent instability of the frameshifting product, and one of the few human cases known to date (CCR5) has been contested recently[16]. Many of the human PRF events were found through homology. A striking example of regulatory conservation is the case of ornithine decarboxylase antizyme (OAZ), which is produced through polyamine-stimulated +1 frameshifting and inhibits polyamine production[11,12]. This negative feedback loop is used by virtually all organisms from yeast to humans to control polyamine levels[17], attesting to the evolutionary success of PRF as a regulatory mechanism.

Frameshifting is generally believed to happen at defined positions consisting of a slippery sequence and a downstream roadblock, most commonly a stable secondary RNA structure like a pseudoknot or an extensive stem-loop structure[2,18]. Most presently known −1 slippery sites follow the pattern X XXY YYZ, with the shift happening from codons XXY and YYZ to XXX and YYY. Some known slippery sites in −1 PRF show divergence from this pattern, e.g., in the NSA2 gene of west nile virus with the slippery site sequence UCCUUUU[19,20]. At +1 frameshifting sites like OAZ ribosomal translocation happens at a very distinct motif (e.g., UCCUGA). Many case studies have contributed to an understanding of the molecular events happening during frameshifting[7,18,21–23], but the general, overarching regulatory principles that determine if and to what extent PRF occurs remain largely unknown.

Here, we developed a massively parallel reporter assay that allows for high-throughput quantification of ribosomal frameshifting in human cells. We designed and tested 13,390 oligonucleotides containing rationally designed variants of known frameshifting signals. We systematically deciphered determinants of PRF efficiency across frameshifting events and assayed natural variation in HIV gag-pol frameshifting, providing the first systematic large-scale investigation of ribosomal frameshifting.

## Results

### A massively parallel reporter assay for PRF
To assay PRF in a comprehensive manner, we designed a synthetic oligonucleotide library containing 12,809 variants with systematic sequence manipulations of previously reported PRF sites (Supplementary Data 1, main set), and 581 sequences of gag-pol frameshifting sites in HIV clinical isolates (collated from http://www.hiv.lanl.gov/, see "Methods" for selection criteria of HIV variants; Fig. 1a).

The oligonucleotides comprising library-specific common primers, a unique barcode and a 162 nt long variable region containing a potential frameshifting site (Fig. 1a, bottom) were synthesized on an Agilent microarray, amplified and cloned in between *mcherry* and *gfp* coding sequences, such that the *gfp* coding frame was shifted by +1 or −1 relative to the original, mCherry-encoding frame (Fig. 1b). If the corresponding frameshift occurs, GFP is made into protein, and GFP fluorescence intensity thus serves as a measure for frameshifting efficiency. We introduced this construct in the AAVS1 locus in the human K562 cell line using zinc finger nucleases (ZFNs), such that every cell has one frameshifting reporter construct from the library and all the variants have the same genomic environment ("Methods" section). For both, the −1 and +1 reporter libraries, we selected a narrow mCherry-positive population to minimize effects coming from the influence of the variable region on overall expression levels (Supplementary Fig. 1a, b, "Methods" section). We sorted this population corresponding to a single integration of the reporter transgene, using flow cytometry into 16 bins according to their GFP fluorescence intensity (Supplementary Fig. 1c, d), and sequenced genomic DNA from all the bins to determine the distribution of each variant across bins. We previously demonstrated that similar approaches are highly accurate and reproducible[24–27], and the consistent bin profiles for many barcode control groups with identical variable region (Supplementary Fig. 1e, "Methods" section) corroborate the low technical noise we are able to achieve.

The distribution of mean GFP expression of all library variants showed a clear peak corresponding to background green fluorescence (Supplementary Fig. 2a, b). We set the lowest mean GFP fluorescence we observed to 0% and the highest—coming from variants with GFP in frame with mCherry—to 100%. Accordingly, we assigned a percentage to every variant that passed filtering for read number, bin profile, and expression levels (by gating for a narrow range of mCherry fluorescence to minimize effects coming from the influence of the variable region on overall expression levels, "Methods" section). This percentage value does not denote the precise rate of frameshifting events, but gives us a meaningful measure of frameshifting efficiencies across PRF events. We thereby obtained measurements for 8972 (67%) and 5922 (44%) sequences cloned into the −1 PRF and +1 PRF reporter, respectively (see Supplementary Data 2 for information and readouts for all library variants).

To test the possibility that the signal in our assay comes from internal translation initiation downstream of *mCherry* (which might lead to GFP expression independently of ribosomal frameshifting in between the *mCherry* and *gfp* coding region), we cloned our library in a modified reporter construct containing a stop codon after the *mCherry* coding region (Supplementary Fig. 2c, top). GFP fluorescence coming from a frameshifting event would in this case be lost. We subjected this library to the same experimental pipeline and obtained green fluorescence measurements for 10,561 variants (79% of the library, 82 and 83% of variants for that we obtained −1 and +1 PRF readouts, respectively; Supplementary Fig. 2c). Green fluorescence was almost exclusively at background levels, showing that the overwhelming majority of tested sequences cannot drive GFP expression on their own.

Another potential source of GFP expression is the removal of cryptic introns in the variable region that would lead to *gfp* being in frame, with *mCherry* in the resulting mature mRNA. To test if this is a source of false positives in our assay, we performed RNA sequencing on the whole library, covering the entire variable region and its surroundings (Supplementary Fig. 2d). We mapped the full-length reads to the library variants to detect splicing events and obtained an RNA readout (at least 100 reads mapped)

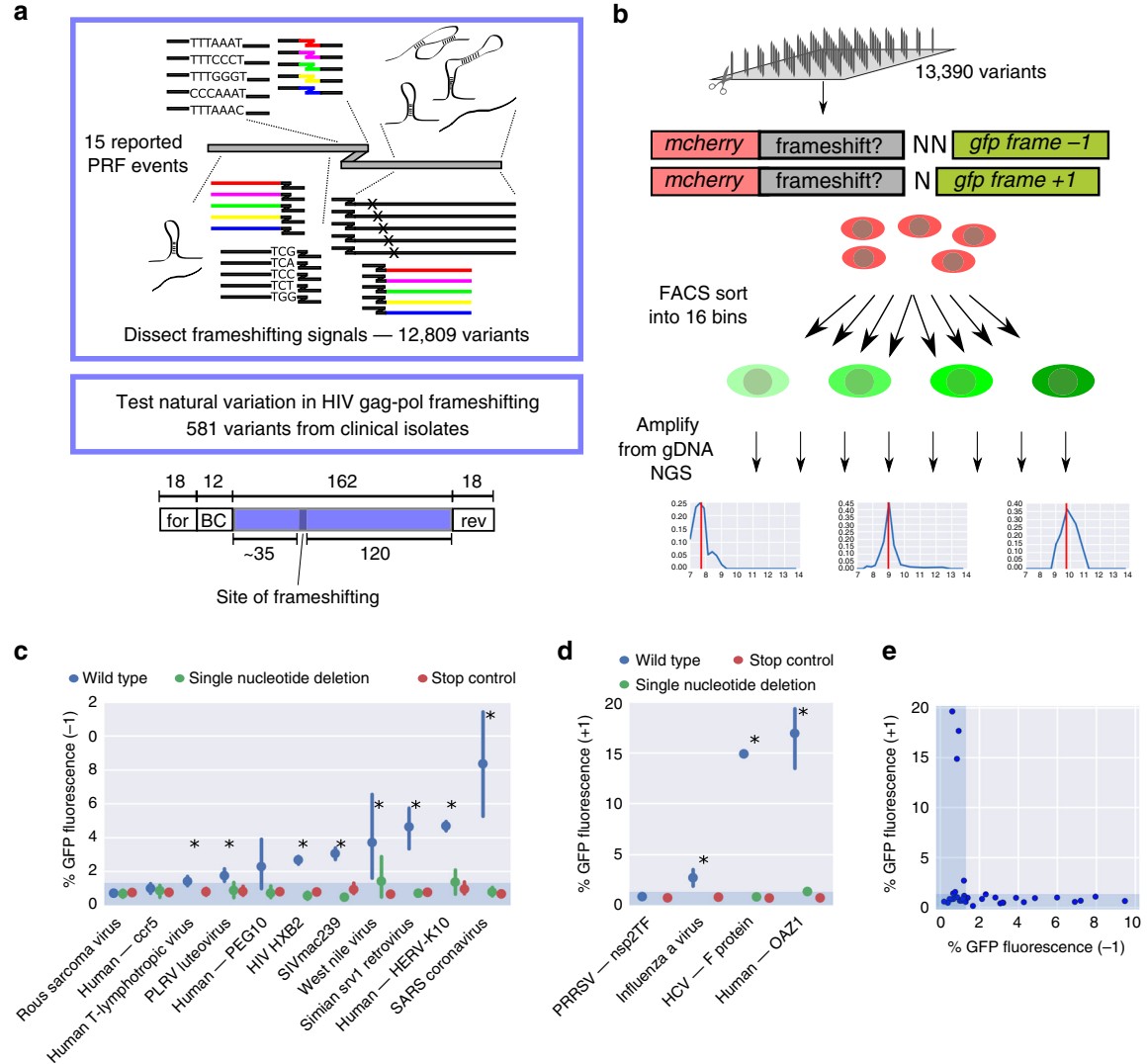

**Fig. 1 A massively parallel reporter assay for programmed ribosomal frameshifting. a** Schematics of the library design; for forward primer, rev reverse primer, BC barcode. **b** Outline of the experimental pipeline; the plots in the bottom constitute representative bin profiles for variants with no (left), low (middle), or intermediate (right) GFP fluorescence. **c**, **d** Mean ± 95% CI of −1 (**c**) and +1 (**d**) PRF signal (% GFP fluorescence) for barcode control groups corresponding to the indicated wild-type PRF sequences (blue), carrying single nucleotide deletions (green) or having a stop codon inserted upstream of the frameshift sequence (red); $n = 22, 2, 13, 32, 10, 25, 21, 8, 13, 24, 9$ (**c**) and $n = 2, 2, 1, 10$ (**d**) wild-type sequences (blue), and $n = 32, 7, 12, 33, 28, 23, 23, 21, 14, 23, 40$ (**c**) and $11, 7, 19, 20$ (**d**) sequences with deletion or stop codon (green and red) tested, in the order they appear on the graph; the shaded area denotes the range of background fluorescence; asterisks denote significant differences between the wild type and the other groups (single nucleotide deletion and stop control) combined (Mann–Whitney $U$ test). **e** Comparison of −1 and +1 PRF reporter readout (% GFP fluorescence) for previously reported PRF sites (cf. **c**, **d**, Supplementary Fig. 3a, c).

for 10,666 variants (80% of the library, 94% of the variants with readout in the −1 frame, and 96% of the variants with readout in the +1 frame). Even without any filtering, only 2.95% of the variants showed gaps in the mapping of at least 20 nt that could represent events of cryptic splicing (Supplementary Fig. 2e, f; 0.95% introducing a shift to frame +1 and 0.64% to frame −1). Missing stretches can also come from synthesis or cloning errors (and are therefore present and filtered out on the DNA level). Considering only those cases that had a potential (degenerate, cryptic) donor or acceptor splice site in the area of the gap, further reduced the fraction of variants for that we cannot rule out a relevant splicing event to 36 (0.55%; Supplementary Fig. 2e, f, red; 0.22% introducing a shift to frame +1 and 0.14% to frame −1). Importantly, even for these cases there was no correlation between the fraction of potentially spliced reads and GFP expression (Pearson $r = 0.01$, $p = 0.21$ for −1 PRF and $r = 0.02$,

$p = 0.09$ for +1 PRF), indicating that cryptic splicing events do not constitute a common source for false-positive signal in our assay.

During oligonucleotide synthesis, errors can occur that can introduce a point mutation or an insertion/deletion. For assaying ribosomal frameshifting, especially the latter type would be detrimental, and therefore we perform full-length DNA sequencing in our assay and consider only reads that exhibit perfect alignment along the length of the variable region covered by sequencing reads (~20 nt in the center of the variable region cannot be covered due to the read length (150 nt from either side)). These synthesis errors, however, also further enrich our synthetic library, as they add additional variants to the collection of sequences tested. Especially single nucleotide deletions or insertions constitute a great internal control, as they differ only in one position from the designed variant. Cases where we obtained

enough reads (after filtering, "Methods" section) for the intended variant (Supplementary Fig. 2g, top, wild type), a single nucleotide deletion abolishing the signal coming from −1 PRF (Supplementary Fig. 2g, middle), and a single nucleotide insertion leading to *gfp* being in frame (Supplementary Fig. 2G, bottom) constitute a proof of principle. They also provide additional evidence that the signal that we measure stems from a (reading frame-sensitive) ribosomal frameshifting event. Looking at the distribution of all single nucleotide deletion and insertion variants from the library reveals the expected peaks at minimal and maximal GFP expression (Supplementary Fig. 2h, i), with a small group of variants being misclassified due to the fact that our reads cover only ~90% of the variable region.

Based on the distribution of all library variants with a stop codon cassette integrated immediately after the *mcherry* coding sequence, we assigned a noise threshold (95th percentile of the stop-vector library distribution, corresponding to 1.3% of the maximal GFP fluorescence; Supplementary Fig. 2c) imposed by autofluorescence of the cells. The contribution of autofluorescence coming from cellular components and metabolites, such as NADH and flavins is negligible when methods for signal amplification are used (e.g., in Luciferase assays commonly used for the quantification of PRF). It does, however, constitute a limitation for detection of GFP fusion proteins made from a single gene copy, and present at low copy number in the cell owing to the fact that the PRF event responsible for its expression happens in only a few percent of translation events. This entails a higher detection limit of our assay compared to, for example, Luciferase assays, which is a necessary compromise to enable high-throughput testing.

Based on the distribution of GFP intensities of variants with a mutation leading to GFP being in frame with mCherry (Supplementary Fig. 2i), we also excluded variants exhibiting a dominant peak with mean GFP fluorescence >2 (ref. [12]; 25% of maximal GFP fluorescence). This value corresponds to the fifth percentile of the distribution of variants with a single nucleotide insertion (above threshold). We thereby aim to rule out biases stemming from DNA frameshifts occurring as synthesis or cloning errors that our mapping strategy might have missed ("Methods" section).

We tested previously reported frameshifting sites (with or without experimental validation, Supplementary Data 1, based on Moon et al.[28] and our own survey of the literature), with multiple different barcodes in our assay and could reproducibly detect GFP fluorescence in the expected frame for many of the previously reported PRF events (Fig. 1c, d, Supplementary Fig. 3a–d). For example, the previously reported PRF sites in HIV-1 (ref. [29]), SARS coronavirus[30], HERV-K10 (refs. [31,32]), simian retrovirus[33], SIVmac239 (refs. [34,35]), human T cell lymphotropic virus[36], and OAZ[11,12] yielded a fluorescent signal between 1.4 and 18% of maximal GFP fluorescence detected in our assay. These values do not necessarily denote the percentage of translation events, in which frameshifting happens, in absolute terms, and are not directly comparable with percentages reported in other studies.

PRF events tested in our assay typically gave a signal over threshold only in either the −1 or +1 PRF reporter, but not in both (Fig. 1e). In cases where the native sequence contained stop codons in any frame downstream of the frameshifting site, we also tested variants with all stop codons mutated ("Methods" section). These typically either did not affect or led to a substantially reduced frameshifting signal (Supplementary Fig. 4a). Moving the slippery site by one to three nucleotides typically abolished the signal, e.g., in the case of SARS (Supplementary Fig. 4b, "Methods" section).

Some previously reported PRF events failed to yield a signal above threshold in our assay. Several reasons like sensitivity to the

larger context of the reporter construct or a general effect of the tested sequence on expression could be possible explanations. In the case of the PRF site in Rous sarcoma virus[37] (Supplementary Fig. 4c), a construct containing 124 nt of sequence downstream to the position of frameshifting was able to induce frameshifting, but shorter versions were not[38]. A sequence motif (termed PK4) at the end of this downstream region was shown to be important for maintaining wild-type frameshifting ability, potentially by creating a pseudoknot-like structure through base-pairing with a complementary sequence in the loop of the main stem-loop structure[39,40]. Our construct (including 120 nt of downstream sequence) does contain the PK4 motif, but it is located at the very 3′ end of the native sequence included in the reporter construct. The sequence context of PK4 was shown to be important for proper folding[39], and therefore the specific context created by our reporter construct might explain the lack of signal.

We also failed to detect a frameshifting signal for CCR5 (ref. [7]; Supplementary Fig. 4c). In addition to potential sensitivity to the larger context of the reporter construct and a specific miRNA that is required for maximal frameshifting efficiency[7], the lack of signal might be due to a potential effect of the PRF site on overall expression levels on the RNA or protein level, as has been suggested for cellular frameshifting sites in general[1]. We select for a narrow range of mCherry levels and therefore expression levels ("Methods" section, Supplementary Fig. 1a, b), which leads to filtering out variants that strongly affect overall expression levels. In the case of CCR5, only 29% of variants with the wild-type sequence passed filtering (Supplementary Fig. 4d; as opposed to 45–97% for other PRF events), which could hint at an effect of the included sequence on expression at the RNA or protein level. Estimating steady-state RNA levels did not show systematic differences between PRF events, yielding a signal above threshold in our assay and those that failed to do so (Supplementary Fig. 4e), but indicated that specifically RNA from variants containing the PRRSV PRF site was present at reduced levels.

Some of the frameshifting signals tested showed variability between variants with identical variable region, but different barcodes (Supplementary Fig. 4c, right). In the case of west nile virus, the majority of sequences tested resulted in a fluorescent signal above threshold, leading to it being classified as frameshifting in our assay ($p = 0.027$ for the difference between wild type and stop control; Fig. 1c); in the case of PEG10 (refs. [8,41]), the majority of sequences failed to yield a signal (after filtering). This points at an instability of the signal in this context, and therefore our assay might not be suitable for investigating these specific events. PEG10 is thought to have a relatively extensive downstream region necessary for frameshifting (~80 nt according to estimates from in vitro testing of truncated regions[41]). Our library includes 120 nt of downstream sequence, but the more extensive the downstream sequence, the higher the sensibility to the context and experimental conditions might be. This is in line with our observation for SARS and HERV-K10, for example (see below). In addition, the required context might be larger in the in vivo situation compared to in vitro.

This instability of the signal might be affected by small sequence changes: for some library sequences measurements of a large number of variants with sequence alterations introduced during synthesis and cloning are available. In the case of PEG10, all the synthesis variants containing a specific mutation (C-to-G at position 7 after the slippery site) show a −1 PRF signal above background (Supplementary Fig. 4f). Likewise, a fraction of the designed sequence alterations introduced in the PEG10 context triggered frameshifting (Supplementary Fig. 4g), especially mutations in the upstream region (Supplementary Fig. 4g left). This shows that minimal sequence changes can have far-reaching (and consistent) consequences for the potential of a sequence to

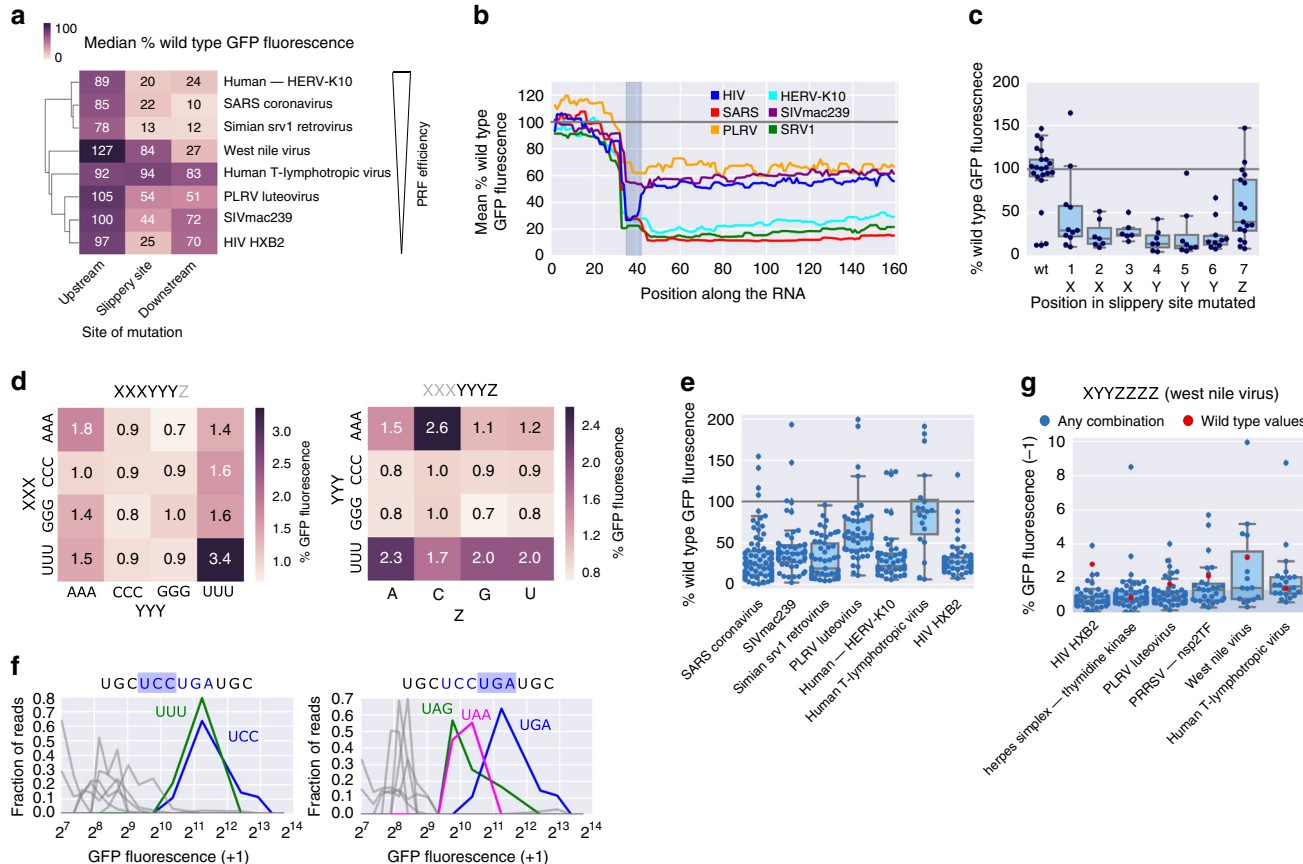

**Fig. 2 Slippery sites are optimized for the respective frameshifting event. a** Clustered heat map of median percent wild-type frameshifting of variants, in which the indicated region (upstream, slippery site (including the preceding codon) and downstream) differs from the native one (n = 37–325). **b** The median percent wild-type PRF signal of variants, in which the indicated position is changed is plotted for the entire length of the variable region; gray box: slippery site. **c** Percent wild-type −1 PRF signal for variants with a point mutation only in the indicated position of the slippery site; the box shows the quartiles of the dataset, while the whiskers show the rest of the distribution except for outliers (n = 22, 11, 7, 6, 7, 8, 10, and 17 sequences tested, in the order they appear on the graph). **d** Heat map showing median percent GFP fluorescence conferred by replacing the native slippery site in −1 PRF sites tested positive in the assay (Fig. 1c), with the indicated combination of slippery site elements. **e** Distribution of percent wild-type GFP fluorescence of variants, in which the slippery site was replaced by any possible combination of bases following the pattern XXXYYYZ; the box shows the quartiles of the dataset, while the whiskers show the rest of the distribution except for outliers (n = 80, 49, 57, 48, 42, 19, and 56 sequences tested, in the order they appear on the graph). **f** Distribution of normalized reads across GFP (+1 frame) expression bins for variants of the OAZ1 frameshifting site, in which either the first (left) or the second (right) codon is replaced; blue lines: wild type; green and magenta lines: bin profile for the variant, in which the native codon was replaced by the indicated one; gray lines: bin profiles for other codons at this position. **g** Percent GFP fluorescence for variants, in which the slippery site was replaced with all possible combinations of the slippery site pattern found in west nile virus (XYYZZZZ); the box shows the quartiles of the dataset, while the whiskers show the rest of the distribution except for outliers (n = 54, 56, 51, 24, 16, and 22 sequences tested, in the order they appear on the graph).

yield a frameshifting signal in the context of our high-throughput assay and probably in any experimental setup.

**Slippery sites are optimized for the respective PRF event.** While previous investigations of frameshifting focused on individual examples, we aimed to identify commonalities and differences between frameshifting sites. To this end, we introduced systematic sequence alterations (such as point mutations, synonymous substitutions, altering and mimicking endogenous secondary structure, etc.; Fig. 1a; "Methods" section) in all contexts and measured the effect on frameshifting in large scale, using our FACS-based assay. Individual PRF events exhibited characteristic sensitivities to sequence alterations (Fig. 2a). The HIV-1 gag-pol PRF site[29] was particularly sensitive to changes in or immediately around the slippery site (any sequence change in this region led to a median PRF signal of 25% of the wild type), while changes in the upstream or downstream region led to PRF signal of—on average—97% (upstream) and 70% (downstream;

Fig. 2a, Supplementary Fig. 5a) of the corresponding wild type. A similar pattern could be observed for SIVmac239. The HERV-K10 (refs. [31,32]), SARS[30], and SRV1 (ref. [33]) PRF sites were highly sensitive to changes in both the slippery site and the downstream region, and in the case of HERV-K10 and SRV1 also showed a significant (p = 0.0004 and 0.0023, respectively; Wilcoxon signed-rank test) reduction following changes in the upstream region (Fig. 2a, Supplementary Fig. 5b). It is noteworthy that the clustering of effects (although done on percent of wild-type PRF signal) resembles the wild-type PRF efficiencies (Fig. 1c), with the most efficient sites (HERV-K10, SARS, and SRV1) showing the highest and broadest sensitivity to sequence alterations.

Comparing per position the median PRF signal for variants, in which this position was changed reveals a sensitivity profile around the PRF site (Fig. 2b, Supplementary Fig. 5c), recapitulating these context-specific characteristics. Across all −1 PRF events tested positive in our assay, not only mutations at the slippery site (marked in gray) and downstream of it, but also

mutations immediately upstream of the slippery site negatively affect PRF efficiency. Other −1 PRF events reported in the literature, but not yielding a signal above background in our assay, tend to show substantially increased signal upon mutations in the areas upstream (PEG10) and downstream (PEG10, CCR5, and Rous sarcoma virus) of the slippery site (Supplementary Fig. 5d). Like the specific examples of frameshifting PEG10 variants described above (Supplementary Fig. 4e, f), this points at a strong sensitivity of these PRF events to the context, in which they are tested.

Most single-point mutations within the slippery sequence abrogated frameshifting and resulted in green fluorescence at background levels, both for −1 and +1 frameshifting sites (Fig. 2c, Supplementary Fig. 5e), demonstrating that we indeed measure frameshifting at the expected site. Replacing the last position in the canonical slippery site X XXY YYZ had—on average—a less severe effect on PRF efficiency, due to the fact that it is the wobble position in the original frame and therefore has larger flexibility (Fig. 2c). In high efficiency PRF sites also this position showed strong and PRF site-specific preferences: in the case of SRV1 (GGGAAAC) replacing the Z position with other options than the endogenous C led to a drastic decrease (U, A) or complete loss (G) of −1 PRF signal (Supplementary Fig. 4e), while in the case of SARS (UUUAAAU) all other nucleotides apart from the endogenous U at the Z position almost completely abolished the signal. To determine whether there is a universally optimal slippery site, we replaced the slippery sequence in our set of −1 PRF events with all possible variations of the pattern XXXYYYZ (Fig. 2d, Supplementary Fig. 4f). This revealed preferences for specific combinations common in known PRF sites (Fig. 2d), like UUUUUUZ (HIV and SIVmac239) and XXXAAAC (HERV-K10, HTLV, PEG10, and SARS), but no universally optimal combination of bases that would lead to maximal frameshifting efficiency and consequently GFP fluorescence across contexts (Supplementary Fig. 5f). Notably, PRF sites showed a high degree of optimization when it comes to the identity of the bases, with any substitution (although preserving the canonical XXXYYYZ pattern) typically leading to a more or less severe drop in PRF signal (Fig. 2e, Supplementary Fig. 5e, f).

Replacing the first or second codon of the OAZ slippery site (UCC UGA) revealed a very limited tolerance to sequence alterations, consistent with earlier results[11]. Replacing the second codon (UGA), the only variants leading to a +1 PRF signal over background where the other two stop codons (UAA and UAG; Fig. 2f, right), reaching ~50% of wild-type frameshifting rates, confirming that a translation termination signal is required for frameshifting[12]. At the first codon position, UUU was the only codon leading to +1 PRF signal similar to wild type (Fig. 2f, left), maybe due to the creation of a stretch of four Us and thereby facilitating repairing of the tRNA in the +1 frame. Indeed, UUU is the second most widely used codon found in that position in the OAZ frameshifting site in other species, e.g., certain fungi and nematodes[42].

Some reported −1 PRF sites deviate from the canonical slippery site pattern (XXXYYYZ). To test the ability of the alternative patterns to induce frameshifting across sequence contexts, we replaced the slippery site in a set of PRF events with all combinations of nucleotides following the pattern found in west nile virus[43,44] (XYYZZZZ, Fig. 2g). In general, the ability of alternative slippery site patterns to cause frameshifting was limited to their native sequence contexts. In this context, however, several different nucleotide combinations following the respective pattern were tolerated. The only exception to this rule was human T-lymphotropic virus (Fig. 2g, endogenous slippery site UUUAAAC), which gave a robust −1 PRF signal corresponding to wild-type levels across many variants following

the (west nile virus) pattern XYYZZZZ. In summary, our data suggest that in most cases a slippery site not following the canonical pattern requires a specific sequence context in order to trigger frameshifting, in accordance with earlier results comparing west nile virus strains with canonical or noncanonical slippery sites[44].

**Downstream regions affect PRF in a context-dependent manner.** The region downstream of the slippery site is thought to be crucial for PRF, typically because it creates a roadblock for the translating ribosome by folding into a stable secondary structure like a hairpin or a pseudoknot[45–48]. To get an estimate of the size of the relevant downstream region in different PRF events, we replaced native regions with a constant sequence, leaving stretches of different length after the slippery site unchanged (Fig. 3a, top; "Methods" section). While in the case of SARS, almost the entire downstream region included in our reporter construct (120 nt) was crucial for frameshifting, in the HIV and SIVmac239 PRF sites only a comparatively small region (<24 nt) was required to achieve PRF signal >50% of wild type (Fig. 3b). The shortness of the necessary downstream sequence in these cases was striking, but in line with earlier evidence demonstrating that base-pairing at the first three to four positions of the HIV downstream stem-loop structure shows the strongest association with PRF efficiency[49].

To identify specific positions crucial for PRF, we performed scanning mutagenesis of regions ranging from 7 to 36 nt downstream of the slippery site (Fig. 3a, bottom). This revealed large differences in the fraction of relevant downstream positions (Fig. 3c), ranging from 6% of single-point mutations reducing −1 PRF efficiency by more than half in the case of SIVmac239 to 53% in the case of SARS (46% for +1 PRF at the OAZ1 site). The HIV gag-pol PRF site (among others) shows remarkable resilience to point mutations (Fig. 3c, Supplementary Fig. 6a), in line with the high degree of genetic variability in HIV. Specific mutations affecting PRF typically disrupt base-pairing (e.g., positions 11, 12, 29, and 30 in HIV, and positions 8–13 and 18–26 in SRV1; Supplementary Fig. 6a) and therefore change the (predicted) secondary structure. In the case of HIV, only mutations in the middle of the stem are detrimental, suggesting that a reduced stem is still able to drive PRF at wild-type levels, and only forcing a complete change in the structure affects frameshifting.

We tested if we can detect these mutational signatures of secondary structure also on the scale of the entire library, and compared the median PRF signal between all variants of a PRF event, in which this position is predicted to be paired or unpaired (Fig. 3d). Preferences for downstream positions to be paired or unpaired reveal the architecture of structural elements downstream of the PRF site. Groups of frameshifting events show remarkable concordance between these preferences (e.g., HIV and SARS, HERV-K10 and SRV1), but length and position of the optimal downstream secondary structures differ between groups (Fig. 3d, upper vs. lower panel).

Replacing the endogenous downstream region with elements that have different primary sequence, but are predicted to fold into the same secondary structure (Fig. 4a, top) abolished frameshifting in most cases, regardless of whether pseudoknots were included or not (Vienna RNA vs. pKiss to determine the structure that served as the input for antaRNA[50,51]; Fig. 4b). Introducing variants of the SRV1 and the HIV downstream structure (without preserving the original sequence, Fig. 4a, bottom; "Methods" section) had the ability to trigger frameshifting in some cases (Fig. 4c), including ones where the wild-type sequence did not yield a frameshifting signal in the context of our assay (CCR5 and PEG10). Strikingly, events associated with higher wild-type frameshifting rates could not be rescued by

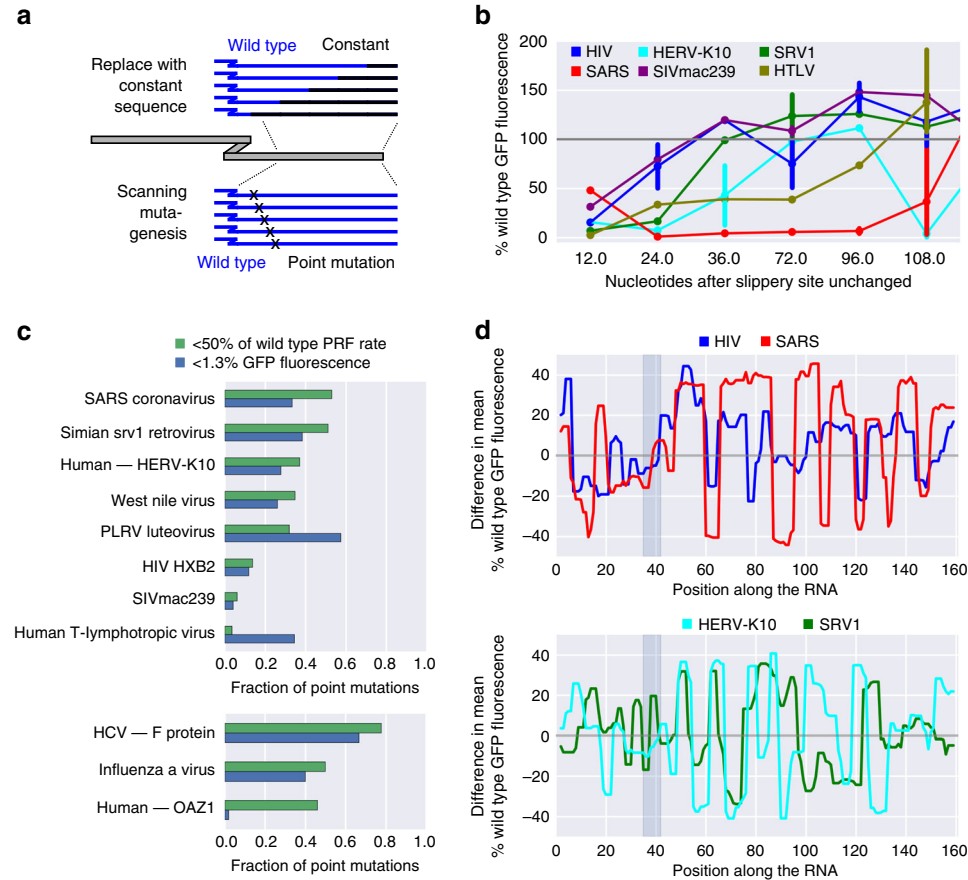

**Fig. 3 PRF sites exhibit different sensitivities to changes in the downstream sequence. a** Schematic of downstream sequence manipulations. **b** Percent wild-type frameshifting rates of variants, in which the native downstream region has been replaced by constant sequences, leaving the indicated number of nucleotides after the slippery site unchanged; $n = 1–3$ per data point and PRF event. **c** Fraction of point mutations in the 40 nt downstream of the slippery site resulting in <50% of wild-type GFP fluorescence (green) and background fluorescence (<1.3% GFP fluorescence, blue), for −1 (top) and +1 (bottom) PRF events; $n = 51, 47, 54, 23, 47, 51, 49, 29$ (upper) and 9, 10, 52 (lower) sequences tested in total. **d** The difference in mean % wild-type frameshifting between variants, in which the indicated position is predicted to be paired vs. unpaired along the variable region; gray box: slippery site.

introducing a secondary structure variant from a different PRF event (Fig. 4d), indicating that more efficient frameshifting sites like SARS seem to be highly optimized, but less tolerant to changes in the type of downstream secondary structure.

Changes in secondary structure (minimum free energy (MFE)) induced by point mutations downstream of the slippery site showed strong correlation with most, but not all frameshifting events (Fig. 4e, left column, Supplementary Fig. 7, left panels). Surprisingly, this correlation between predicted MFE of secondary structure variants of the native downstream region and GFP fluorescence of −1 PRF reporters was lost, when replacing the downstream region with structural variants (more) different from the native sequence (Fig. 4e, middle column, Supplementary Fig. 7, middle panels; these structures generally had the potential to induce frameshifting in certain contexts; Fig. 4c). Likewise when testing for correlation across all variants of a PRF site (Fig. 4e, right column, Supplementary Fig. 7, right panels), there was no association of lower MFE with higher PRF rates.

Many stimulatory downstream sequences are thought to fold into a pseudoknot (e.g., SRV1 (refs. [33,52]), west nile virus[19], and coronaviruses[46,53]), a more complex folding pattern that is not covered by many secondary structure predictions, including the Vienna RNA package[54,55]. We therefore repeated the analyses using pKiss[56], an algorithm that can detect certain classes of pseudoknots (see Supplementary Data 3 for pKiss-predicted structures of library variants with systematic sequence changes in

the downstream region). MFE calculated using Vienna RNAfold and pKiss, respectively, show good correlation (Supplementary Fig. 6b). We repeated the above analysis and observed the same pattern as with RNAfold, namely widespread association between secondary structure (low MFE) and higher PRF signal for small deviations from the native sequence (Fig. 4f, left column, Supplementary Fig. 8, left panels), but not otherwise (Fig. 4f, middle and right column, Supplementary Fig. 8, middle and right panels). This is in line with the fact that we do not see systematic differences between PRF events whose downstream sequence was shown to fold into a hairpin structure (e.g., HIV-1 (refs. [47,57]) and SIVmac239 (ref. [34])) or a pseudoknot (e.g., SRV1 (ref. [33,52]) and SARS[46,53]).

This observation does not imply a (counterintuitive) tendency for more open secondary structure to promote frameshifting. As we do not find evidence for an effect of MFE on overall RNA levels (Supplementary Figs. 7–9; e.g., through triggering No-Go decay[58]), we suggest that this reflects a strong preference for downstream secondary structure rigidity to be close to that of the wild-type sequence (in particular for SARS, SRV1, and HERV-K10, Supplementary Figs. 7 and 8, right panels). Although present secondary structure prediction algorithms might not be sufficiently accurate for this task, especially in the case of pseudoknots, our results support a view according to which not only the structure, but also the sequence downstream of a slippery site is critical for triggering frameshifting.

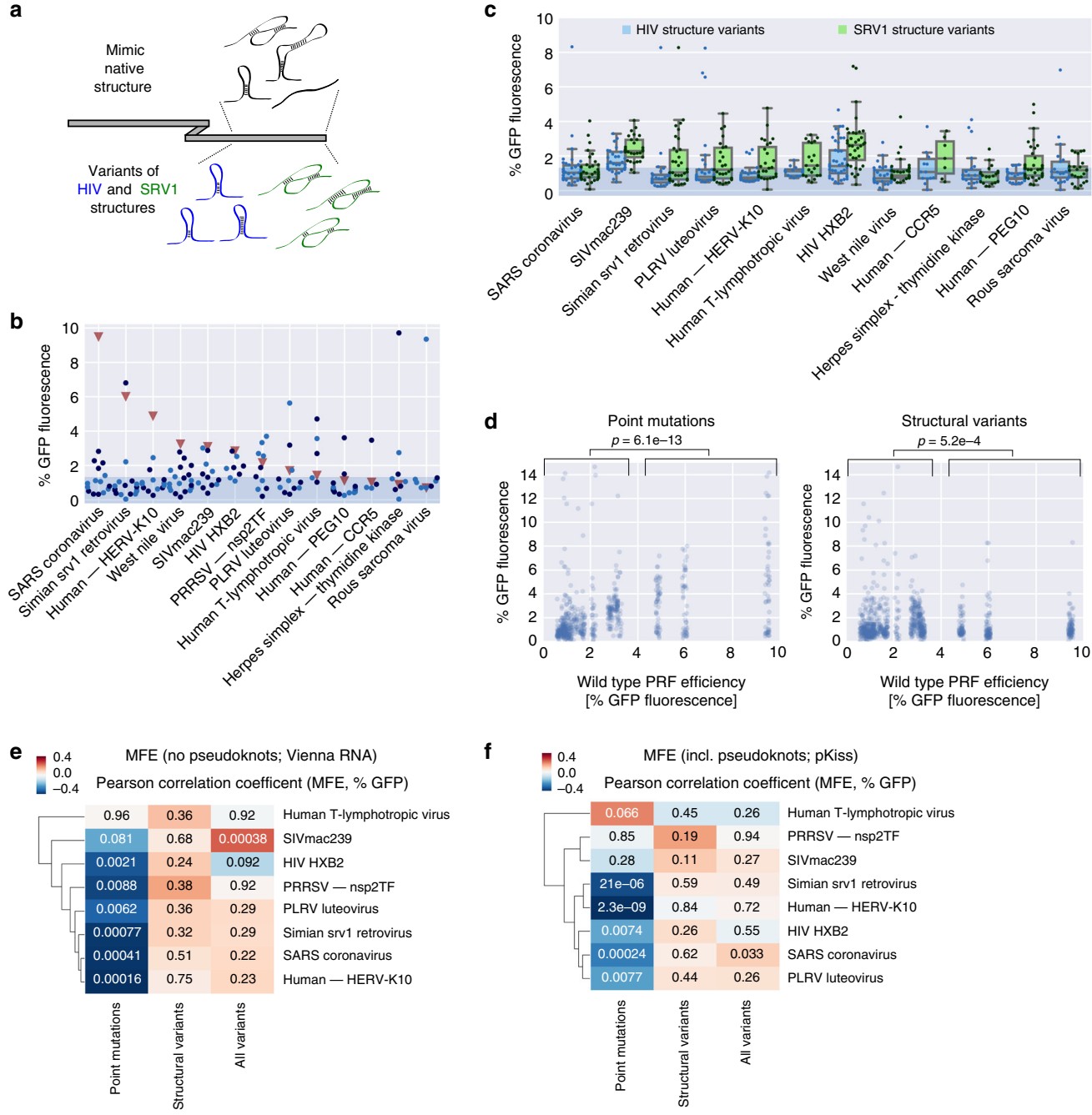

**Fig. 4 Downstream RNA structure affects PRF efficiency in a context-dependent manner. a** Schematic of downstream structure variations. **b** Percent −1 GFP fluorescence of variants, in which the downstream region is replaced by a synthetic sequence predicted to fold into the respective native secondary structure (using Vienna RNA (only stem-loop; light blue) or pKiss (including pseudoknots; dark blue; Methods section)); the shaded area denotes the range of background fluorescence; n = 15, 17, 13, 19, 12, 8, 11, 11, 5, 17, 4, 8, and 8 sequences tested, in the order they appear on the graph. **c** % −1 GFP fluorescence of variants, in which the downstream region is replaced by a synthetic sequence predicted to fold into the HIV (blue) or SRV1 (green) secondary structure (or variations thereof; Methods section); the box shows the quartiles of the dataset, while the whiskers show the rest of the distribution except for outliers; the shaded area denotes the range of background fluorescence; n = 38/37, 32/29, 30/34, 32/29, 32/27, 16/21, 41/36, 34/31, 14/7, 23/20, 37/35, and 25/25 sequences tested for HIV and SRV1 structure variants, respectively, in the order they appear on the graph. **d** % GFP fluorescence of variants, in which the downstream region carried a single-point mutation (left) or was replaced with sequences predicted to fold into variations of PRF-inducing structures, plotted against the corresponding wild-type values; p-values (Mann–Whitney U) for comparisons between groups of low (<4% GFP fluorescence) and high wild-type PRF (>4% GFP fluorescence). **e**, **f** Clustered heat map of Pearson correlation coefficients between % −1 GFP fluorescence and minimum free energy of regions after the slippery site (computed using Vienna RNAfold on the 40 nt following the slippery site (**e**) or pKiss on the 120 nt following the slippery site (**f**)) for groups of variants of the indicated PRF sites; values denote the p-values of the correlations.

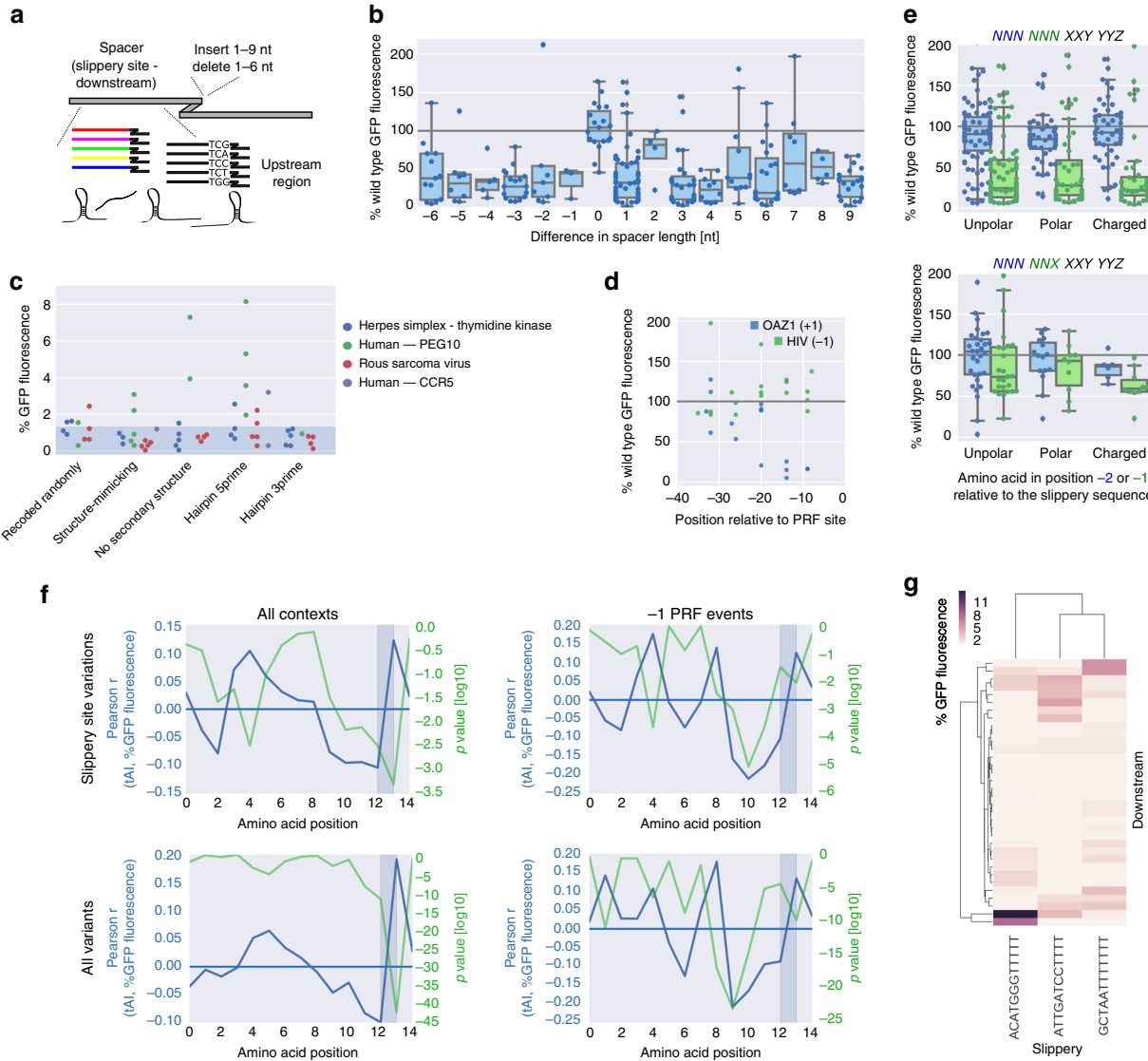

**Fig. 5 Additional regulatory elements contribute to PRF. a** Schematic of sequence manipulations upstream and immediately downstream of the slippery site. **b** Boxplots of % wild-type frameshifting rates for variants of −1 PRF sites, in which the indicated number of nucleotides were deleted (negative numbers) or inserted (positive numbers, taken from two different constant sequences) after the slippery site; the box shows the quartiles of the dataset, while the whiskers show the rest of the distribution except for outliers; n = 13, 7, 4, 17, 7, 3, 16, 54, 4, 27, 10, 12, 23, 8, 6, and 23 sequences tested, in the order they appear on the graph. **c** % GFP fluorescence for variants of −1 PRF sites with the sequence upstream of the slippery site being randomly recoded or replaced with sequences predicted to have the indicated secondary structure; the shaded area denotes the range of background fluorescence. **d** % wild-type frameshifting rates (+1 for OAZ, blue, and −1 for HIV, green) of variants, in which the native upstream region has been replaced by constant sequences up to the indicated position relative to the PRF site. **e** Boxplot of percent of wild-type frameshifting rates for variants, in which the −2 or −1 amino acid relative to the slippery site is replaced with an amino acid from the indicated groups, for codons which would maintain the slippery site pattern XXXYYYZ (bottom) or not (top); the box shows the quartiles of the dataset, while the whiskers show the rest of the distribution except for outliers; n = 95/79, 43/39, 47/37 and 31/25, 16/10, 6/10 sequences tested. **f** Pearson correlation coefficient (blue) and associated two-tailed p-values (green) between tAI at the indicated position of the original reading frame and % GFP fluorescence. **g** Clustered heat map showing all possible combinations of 3 synthetic slippery sites, 34 synthetic downstream variants, and up to 5 synthetic upstream regions (minimal value 1.3%).

**Additional regulatory elements contribute to PRF.** The distance between the slippery site and the downstream secondary structure has been shown to be important for efficient frameshifting[46,59,60], probably to position the paused ribosome at the slippery site[45]. To test whether this is generally true or if there are PRF event-specific differences, we introduced between one and nine nucleotides from different constant sequences or deleted between one and six, mutating downstream stop codons as required (Fig. 5a top). Most insertions or deletions led to substantially reduced frameshifting signal (Fig. 5b), no matter what the context and if the original downstream coding frame was preserved or not, suggesting that the exact position of the downstream regulatory elements is important across PRF events. This is in contrast to an earlier study reporting unchanged or even dramatically increased −1 frameshifting efficiency at the HIV site upon reducing spacer lengths, which the authors attribute to a change in local secondary structure triggered by the specific deletion introduced in their construct[49]. Overall, our results are in line with the regulatory elements being in optimal distance from one another, and the downstream secondary structure creating a roadblock that causes the ribosome to pause exactly at the slippery site[45,48].

While mutations in the slippery site and the downstream region typically have a negative effect on frameshifting efficiency (Fig. 2a), changes in the upstream region have the potential to lead to higher −1 PRF signal (e.g., PEG10, Supplementary Fig. 5d). This suggests that inhibitory signals upstream of the frameshifting site as found for SARS[61] could be a more widespread property of PRF sites, probably creating a balance to ensure that frameshift-promoting signals like a rigid downstream secondary structure do not lead to a complete inhibition of translation and potentially degradation of the mRNA. To test for effects of upstream secondary structure on PRF, we replaced the region upstream of the slippery site with a recoded version (encoding the same amino acid sequence), a sequence predicted to fold into the same secondary structure as the wild type, a sequence lacking any strong predictable secondary structure or a hairpin at the 5′ or 3′ end. Specifically in the case of PEG10, a hairpin at the 5′ end of the upstream region or a lack of secondary structure resulted in a frameshifting signal above threshold (2–8% GFP fluorescence; Fig. 5c) as opposed to the wild-type sequence (Fig. 1c, see our discussion of potential reasons above).

The upstream region of OAZ1 is thought to harbor signals enhancing frameshifting[62]. By testing variants in which increasing portions of the upstream region have been replaced with constant sequences, we find these PRF-promoting signals to be located in the 20 nt before the slippery site (Fig. 5d, decrease of PRF signal on average to 31.3% of wild type, $p = 0.0077$, Wilcoxon signed-rank test, effect on region further upstream not significant). In contrast to the other −1 PRF sites, HIV frameshifting rates show an increase upon mutation of the corresponding upstream region (Fig. 5d, mean increase of PRF signal to 128.2% of wild type, $p = 0.013$, Wilcoxon signed-rank test, effect on region further upstream not significant), indicating the presence of an inhibitory element in this region.

We expanded our search for properties affecting PRF efficiency and examined the effect of the codons preceding the slippery site. We found that the presence of a charged amino acid immediately upstream of the slippery site significantly reduced frameshifting signal, even when the sequence of the slippery site was unchanged (Fig. 5e, $p < 0.0015$, Wilcoxon signed-rank test). We examined also the influence of decoding efficiency as measured by the tRNA adaptation index (tAI) on PRF, and detected an increasingly negative correlation between tAI and PRF signal in the 0 frame codons leading up to the site of ribosome slippage (Fig. 5f). This observation might indicate that progressive slowing down of the ribosome by limited tRNA availability might contribute to stalling at the slippery site and frameshifting.

To not limit our analysis to endogenous frameshifting events, we designed completely synthetic regions corresponding to a slippery site, and the upstream and downstream region with different sequence and structural properties. Some fully designed sequences were able to trigger frameshifting and resulted in a −1 PRF signal of up to 11% GFP fluorescence, approximately corresponding to the wild-type frameshifting rate of the most efficient event tested here (SARS coronavirus). We tested all different downstream regions with synthetic slippery sequences resembling common types of −1 PRF sites and found pronounced differences in combinatorial preferences (Fig. 5g), showing that also fully synthetic frameshifting events exhibit the combinatorial preferences and context-dependent peculiarities observed for native PRF events.

### Prediction of frameshifting potential and efficiency. Having accurate quantitative measurements for large collections of frameshifting sites, we aimed to predict frameshifting efficiency, using machine learning approaches. Based on previous studies and our own findings, we used the identity of slippery site positions, tAI of codons around the frameshifting site, amino acid class, MFE, and pairedness of positions downstream of the frameshifting site (see "Methods" section), alone and combined, as features and built computational models based on Gradient Boosting Decision Trees (XGBoost[63]; Fig. 6a, "Methods" section). These models were either estimating whether a sequence has the ability to induce frameshifting (i.e., yield a fluorescent signal above threshold (1.3%, see Supplementary Fig. 2a–c); classification) or quantitatively predicting PRF efficiency (approximated by GFP fluorescence intensity in our assay; regression). We achieved high accuracy (up to an area under the receiver operating characteristic curve (ROC AUC) = 0.93 for the classification of a sequence as yielding a PRF signal or not, and Pearson $r = 0.81$ for the comparison between measured and predicted frameshifting signal (% GFP fluorescence)) when training our model on variants of specific frameshifting events and predicting unseen variants from the same event (Fig. 6b, c). Despite the differences in the regulatory characteristics between individual PRF events, we could predict frameshifting ability (Fig. 6d, Supplementary Fig. 10a, ROC AUC = 0.89; area under the precision-recall curve = 0.8) and frameshifting signal (% GFP fluorescence; Fig. 6e, Pearson $r = 0.53$) also for a pool of all sequence variants tested, no matter whether they yielded a detectable frameshifting signal in our assay or not (Fig. 6b, c, Supplementary Fig. 10a; −1 PRF events vs. all library variants).

To determine to what extent a model trained on one PRF event could predict frameshifting potential and −1 PRF signal in a different context, we trained our model (based on the full set of features) on sequence variants derived from one event and predicted on variants of another event (Fig. 6f, Supplementary Fig. 10b). While performance is usually best for variants of the same PRF event, this approach revealed functional similarities that allowed relatively accurate prediction on other PRF events (ROC AUC up to 0.94, Pearson $r$ up to 0.69). There is considerable overlap with other attempts of functional clustering, e.g., based on sensitivities to mutations (Fig. 2a), with SRV1 and HERV-K10 PRF sites showing similar behavior and HIV variants yielding good prediction scores for models trained on data from any retroviral PRF event (HIV, SIVmac239, and SRV1; Fig. 6f). The SARS PRF event, however, is not amenable to prediction based on models trained on data from other PRF events. Prediction accuracy also reflects the specific components of particular importance for a PRF event, e.g., the slippery site for HIV (Supplementary Fig. 10c) and the downstream region for SRV1 (Supplementary Fig. 10d). In these cases, frameshifting efficiency can be predicted using models trained on data from almost any other PRF event.

Interpretation of the prediction models provides an additional approach to identify the most important properties of a (potential) PRF site. Here, we used Shapely (SHAP) values[64] for determining the contribution of each feature to the prediction result of every sample (Supplementary Fig. 11a). The features of the different subsets driving the prediction provide additional information about the effect of specific sequence properties on the readout (Supplementary Fig. 11a), e.g., recapitulating earlier observations that XXXCCCZ (Y = C) and XXXGGGZ (Y = G) slippery sites are disfavored (Supplementary Fig. 11a, upper left, cf. Fig. 2d). For a model trained on the MFE of different lengths of downstream region, stronger secondary structure not always led to the prediction of higher PRF efficiency (Supplementary Fig. 11a, lower left). When taking into account regions of, e.g., 60 nt after the slippery site (dg60), a lower feature value favored prediction of PRF (Supplementary Fig. 11b). While the positive effect of strong secondary structure (low MFE) in the first 60 nt on PRF prediction was strongly enhanced if also the downstream

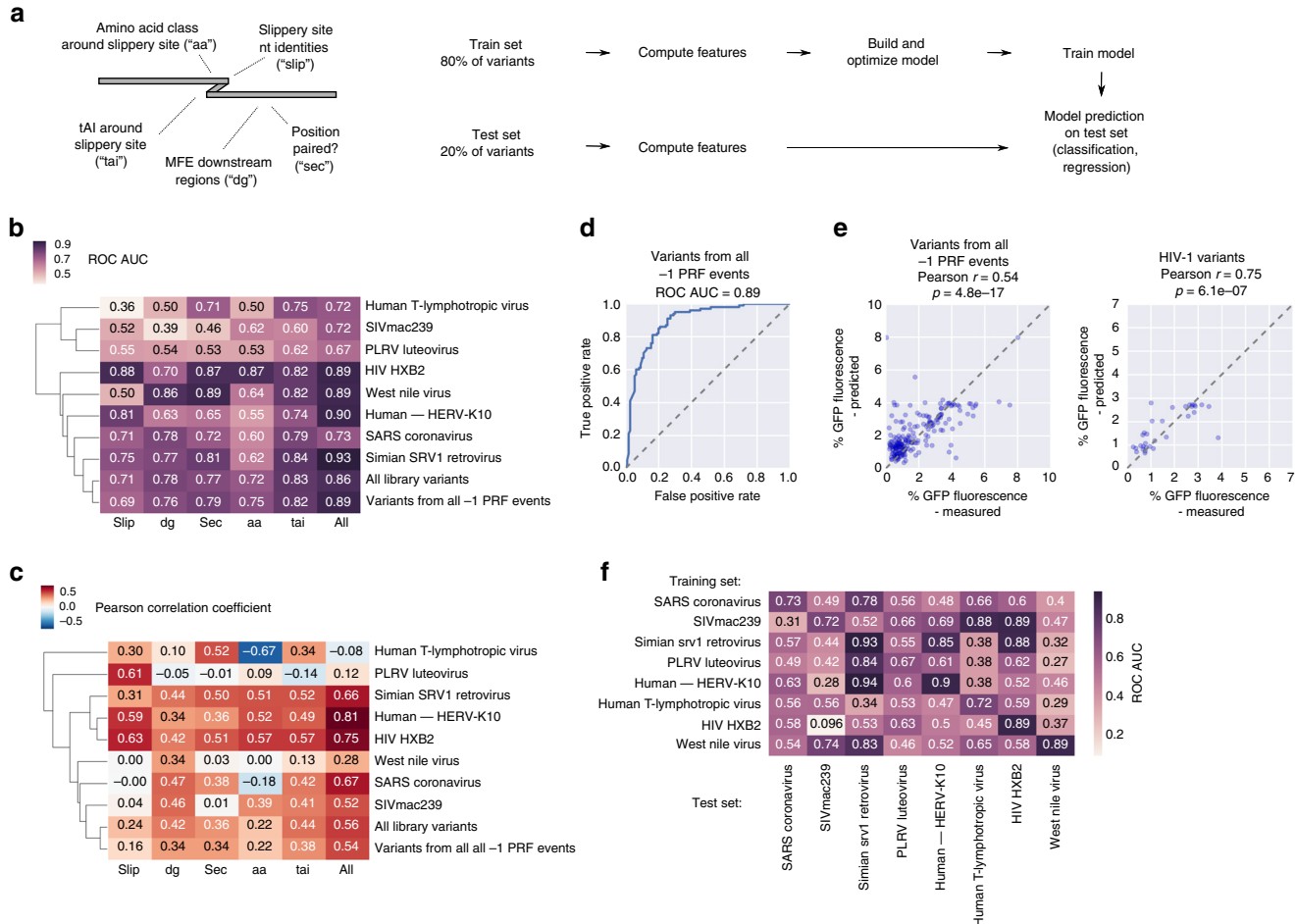

**Fig. 6 Prediction of frameshifting potential and efficiency. a** Outline of the process of generating features and building and testing predictive models. **b**, **c** Prediction scores (area under the receiver operating characteristic curve (ROC AUC) in the case of classification (**b**) and Pearson correlation coefficients in the case of regression (**c**)) on held-out test data for the indicated PRF event(s) or the entire library using different feature sets (slip: nucleotide identities in the canonical XXXYYYZ slippery site; tai: tAI scores around the slippery site, aa: amino acid class (unpolar, polar, and charged) around the slippery site, dg: MFE of downstream regions, sec: predicted pairedness of downstream positions). **d** ROC curve showing performance of a classifier trained on 80% of all designed library variants passing filtering and testing on the remaining 20%. **e** For the test set (20% of all designed library variants of all −1 PRF events (left) or HIV-1 (right) passing filtering), the measured % GFP fluorescence is plotted against the model prediction (trained on 80% of all library variants of all −1 PRF events (left) or HIV-1 (right) passing filtering); Pearson correlation coefficient and the associated two-tailed $p$-value are reported. **f** Prediction scores (ROC AUC) on held-out test data (20%) for the indicated PRF event (columns) using the full set of features and training the model on the training data (80%) from the indicated PRF event (rows).

region as a whole showed low MFE (Supplementary Fig. 11b), in general, we find strong secondary structure of the entire downstream region (120 nt) to be associated with prediction of no PRF (Supplementary Fig. 11a). This could either be explained through an effect on frameshifting efficiency or on RNA stability and overall expression levels. The latter option is unlikely to explain the effect in full, as we do not observe a correlation between MFE of the downstream region and steady-state RNA levels (Supplementary Fig. 11c).

**Assessing the natural variation in HIV PRF efficiency.** The HIV gag-pol frameshifting site is arguably one of the most intensely studied examples of PRF and—due to its critical importance for the viral replication cycle—has been repeatedly suggested as an antiviral drug target[5,6]. To assess the natural variation in frameshifting rates in HIV-1, we assembled a set of 581 sequences from clinical isolates between 1976 and 2014 (http://www.hiv.lanl.gov/, "Methods" section), which differ in the sequence surrounding the frameshifting site, but not in the slippery site itself (Supplementary Fig. 12a).

Frameshifting rates of the variants are distributed around the rate observed for the lab strain HXB2 (Fig. 7a). This constitutes actual differences in frameshifting rates and not only experimental variability, as isolated clones show remarkably good correlation given the small differences in PRF signal we are measuring (Fig. 7b). Secondary structure showed the best correlation with frameshifting rates when considering the first 30 nt after the frameshifting site (Supplementary Fig. 12b), matching the region of high sequence conservation (Supplementary Fig. 12a). We grouped the HIV variants based on subtype and found significant differences between the groups (Fig. 7c, $p < 8 \times 10^{-11}$, one-way ANOVA), most notably higher frameshifting rates in subtype C ($p < 6 \times 10^{-12}$ for the difference between C and B), in contrast to an earlier report[65]. HIV subtypes show distinct geographical distributions[66], and consequently we also observed differences between countries of origin (Supplementary Fig. 12c, $p < 2 \times 10^{-4}$), but no change in frameshifting rates over time (Supplementary Fig. 12d, $p = 0.4$).

Optimal gag-pol frameshifting rates have been proposed to be critical for virulence[67,68]. In order to link the frameshifting rate

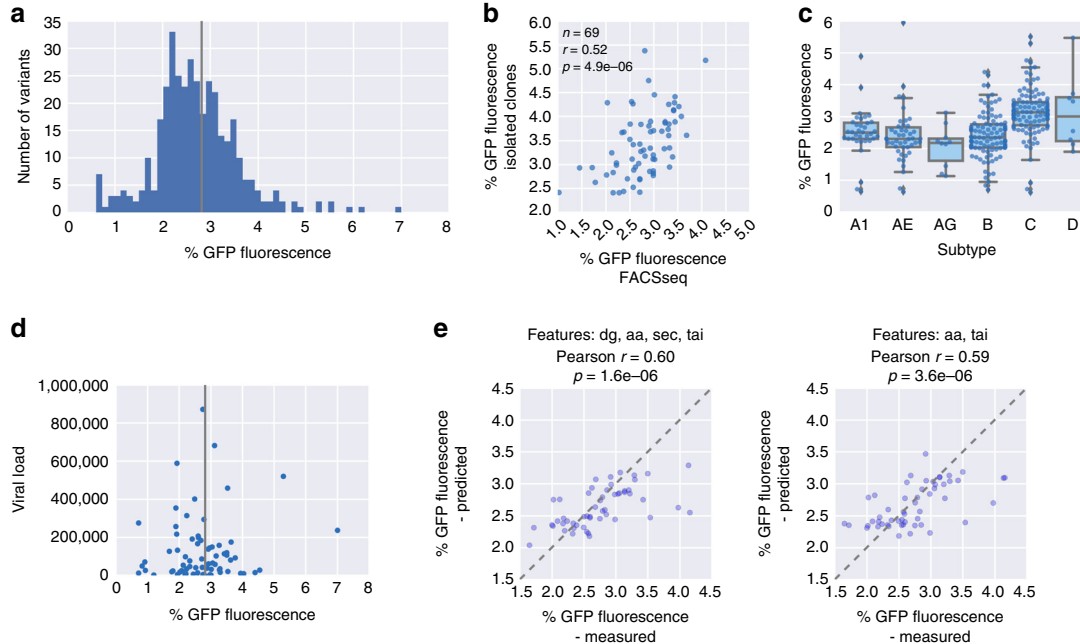

**Fig. 7 Testing of PRF sites from HIV clinical isolates reveals subtype differences and associations with viral load. a** Distribution of percent GFP fluorescence of measured HIV variants, $n = 368$. **b** Percent GFP fluorescence as determined by FACSseq is plotted against the value determined by measuring the corresponding isolated clone by FACS; Pearson correlation coefficient and the associated two-tailed $p$-value are reported. **c** Boxplot showing percent GFP fluorescence of HIV gag-pol PRF variants coming from the indicated subtypes; the box shows the quartiles of the dataset, while the whiskers show the rest of the distribution except for outliers; $n = 35, 46, 10, 103, 112$, and 8 sequences tested, in the order they appear on the graph. **d** Viral load (HIV titer) is plotted against percent GFP fluorescence determined for the natural gag-pol PRF site variant from the corresponding clinical isolate; vertical line: HIV HXB2 wild-type percent GFP fluorescence, $n = 77$. **e** Predicted vs. measured percent GFP fluorescence for 20% of the HIV variants (held-out test set) with the full feature set except for the identity of slippery site positions (left) or after removing MFE features (middle) or all secondary structure derived features (right) features: Pearson correlation coefficient and the associated two-tailed $p$-value are reported.

measured in our reporter assay with the replicative success of the corresponding HIV isolate, we compared the viral load in patients (where available) to frameshifting rates of the causal HIV isolate (Fig. 7d). Although viral load is influenced by many factors, we nevertheless observed a clear trend for patients with high viral load to have frameshifting rates close to "wild type" (the lab strain HXB2). These data present optimality of frameshifting rates as a hallmark of HIV infection and as being associated with infectious success, underscoring the potential for drugs altering the efficiency of gag-pol frameshifting.

In order to predict frameshifting efficiency of naturally occurring variants of the HIV gag-pol frameshifting site, we built a model based on the feature sets described above. Despite eliminating the most important feature of the HIV gag-pol PRF site (the slippery sequence) by restricting our analysis to clinical isolates with no sequence alterations in and around the canonical UUUUUUA and despite the narrow range of frameshifting rates (Fig. 7a), we could accurately predict frameshifting rates of unseen variants (Pearson $r = 0.6$; Fig. 7e, left). Interestingly, removing the features based on pairedness of downstream positions and MFE led to similar agreement between measured and predicted values (Fig. 7e, right). This is in line with the limited importance of the exact downstream structure in comparison to other PRF events tested. Including designed HIV variants in the training set yielded similar prediction accuracy (Supplementary Fig. 12e), indicating that our model can learn the relevant rules from our set of natural variants alone. Taken together, these results show that a model based on data from our assay is sensitive enough to detect even subtle differences between HIV variants and allows prediction of frameshifting rates with an accuracy of clinical relevance.

## Discussion
Here, we combined fluorescent frameshifting reporters with rational design of DNA sequences and high-throughput testing to systematically decipher the rules governing PRF. Our approach aims at a systematic and comparative assessment of the commonalities and peculiarities in the regulation of frameshifting efficiency across PRF events. It is complementary to in-depth analyses of individual frameshifting sites yielding detailed structural or kinetic insights[69–74]. The high-throughput nature of our approach entails particular limitations. While we show that our assay is highly sensitive to changes in the PRF signal (e.g., in the testing of natural isolates of HIV) and therefore able to identify even subtle effects of sequence alterations on frameshifting, our detection limit is higher than in assays using amplification methods and/or overexpression of the reporter. Moreover, many of the general caveats in using reporter systems apply also here, such as the non-native sequence context and expression levels, the effect of the tested sequence on fluorescent readout and protein stability, and the concentration of potential trans-acting proteins or metabolites in the particular cell type used in the experiment.

Prediction of PRF events has up to now been largely limited to identifying sequences matching the canonical pattern for slippery sites, followed by a downstream secondary structure[75,76]. To the best of our knowledge, no attempts at quantitatively predicting the effect of sequence variation exist to date. We used our measurements of frameshifting signal of thousands of variants of known PRF sites to build a computational model based on known and novel sequence features affecting frameshifting, leading to accurate prediction of frameshifting potential (up to ROC AUC = 0.93) and frameshifting rates (up to Pearson $r = 0.81$). In many

cases, sequence properties other than slippery site identity or downstream secondary structure rigidity yielded high prediction scores, emphasizing the importance to include additional features, such as amino acid properties and tRNA availability in the investigation of PRF regulation.

By controlled sequence and structure manipulations in multiple contexts, we aimed to dissect and directly compare the regulatory architecture of 15 frameshifting events, revealing a great diversity in regulatory strategies, involving upstream and downstream sequence and structural elements and amino acid properties. We demonstrate a high degree of optimization and specialization of regulatory mechanisms. For example, no universally optimal slippery site exists, but in each of the PRF events tested the native slippery site was found to be optimal given the sequence context.

A correlation between secondary structure rigidity and PRF efficiency has been shown previously[77–79], while also other properties of secondary structures like their plasticity have been reported to be associated with frameshifting rates[80,81]. Our results corroborate the functional link between MFE of downstream structures and show that this is true for most PRF events tested. However, this correlation generally depended on the length of the downstream region taken into account, in agreement with the specific association between PRF efficiency and thermodynamic stability of the start of the HIV stem-loop structure reported earlier[49]. Furthermore, the correlation between PRF signal and MFE only held true, when testing small variations of the native sequence and structure; when sequences and structures different from the native one were tested, this correlation was lost.

Slippery site and a downstream stimulatory signal are central— and therefore intensely studied—regulatory elements present in all known PRF events. Here, we expanded the repertoire of regulatory inputs and demonstrated that both tRNA availability, as well as amino acid properties can affect PRF efficiencies across native sequence contexts.

Ribosome stalling due to limited tRNA availability has the potential to lead to slippage of the ribosome[82,83] and contributes to nonprogrammed frameshifting in huntingtin[84]. Availability of tRNAs decoding slippery site codons have been shown to influence HIV gag-pol frameshifting efficiency[85], and HIV-1 itself has the ability to modulate the tRNA pool of the host cell[86]. Our observation that limited tRNA availability in the codons leading up to the slippery site progressively contributes to PRF efficiency is in line with these data and expands the notion of an influence of the tRNA pool on frameshifting beyond the slippery site codons and individual PRF events.

In addition, our analyses highlighted amino acid properties around the frameshifting sites as another factor influencing PRF efficiency. Specifically, a charged amino acid at the last position before the slippery site was associated with reduced frameshifting rate. Together with reports of a frameshift-promoting effect of specific amino acid sequences upstream of the frameshifting site in the case of +1 PRF in fungal OAZ genes[87] and the copA gene of *Escherichia coli*[88], this observation provides additional evidence for a role of the nascent peptide in regulating ribosomal frameshifting.

In general, our results reveal functional groups of PRF events that exhibit greater similarity in terms of their sensitivities to mutations in regulatory regions and congruence in the preferences for secondary structure. These functional groups also manifest themselves in a higher prediction accuracy within these groups. The groups resemble similarities also according to other criteria, e.g., in their native secondary structure. PRF events, whose downstream stimulatory sequence has been shown to be a stem-loop (e.g., HIV-1 (refs. [47,57]) and SIVmac239 (ref. [34])) tend to be more resilient and less affected by small sequence changes

than those with a downstream pseudoknot (e.g., SRV1 (ref. [33,52]), as well as SARS and other coronaviruses[46,53]). Typically, these cases exhibit higher fluorescence signal in our assay (Fig. 1c), but greater sensitivities to even small sequence changes (Figs. 2b and 3c, Supplementary Figs. 4a, b and 5a) and requirements for a more extensive downstream region. This might also entail a greater sensitivity to the experimental context and therefore explain the lack of signal (above the noise threshold) for previously reported cases, depending on a pseudoknot as frameshift stimulatory signal like Rous sarcoma virus[37,39]. In other cases, low wild-type rates of frameshifting, requirements for additional trans-acting factors, or an effect on overall expression levels might explain the lack of signal in our assay.

Comparing the HIV gag-pol site with other PRF events shows the remarkable robustness and exceptional tolerance for various non-native downstream stimulatory signals. More than in other cases, maintaining the wild-type slippery site UUUUUA is crucial for frameshifting, and tends to be absolutely invariable across otherwise diverse natural isolates[89]. The quantitative nature of our assay allows us nevertheless to accurately quantify the effect of sequence variation away from the slippery site, revealing subtype-specific differences. Furthermore, we devised a machine learning model that is able to predict the natural variation in HIV frameshifting rates for novel variants (Pearson $r = 0.60$). Precise stoichiometry of frameshifted and non-frameshifted product is crucial for HIV virulence, and the ability to predict frameshifting rates of naturally occurring variants of the gag-pol PRF site therefore has important clinical implications, both in the case of HIV, as well as in the case of PRF events in other viruses.

This high-throughput investigation of PRF also constitutes an addition to the experimental toolbox for studying translational recoding and highlights many starting points for further investigations into the regulatory basis and functional importance of frameshifting. In addition to deciphering the effect of cis-regulatory elements, regulation by trans-acting factors like the recently discovered case of the frameshift repressor shiftless[90] constitutes another area, where a systematic approach like the one presented here can yield novel insights into the mode of action and its specificity. Libraries of frameshifting reporters provide a platform for screening putative modifiers of ribosomal frameshifting, offering a powerful tool to identify ways to generally or selectively control frameshifting, and opening new possibilities for interfering with viral replication and for controlling cellular processes depending on translational frameshifting.

## Methods

**Synthetic library design—general design notes**. Oligonucleotides were designed to maintain a constant length of 210 nt. Restriction sites used for cloning were excluded from the design. All the variants were composed of an 18 nt forward primer, 12 nt barcode sequence, 162 nt variable region, and 18 nt reverse primer sequences. DNA barcodes were designed to differ from any other barcode in the library in at least 3 nt.

**Synthetic library design—selection of frameshifting sites**. Sixty PRF events reported previously or listed in FSDB[28] were included in the library design. Sequences were aligned in a way that the site of frameshifting would be at the same position in all variants (slippery site ending at position 42). In case there were stop codons downstream of the PRF site in any of the three frames, additional versions with stop codons mutated (changed to TGG) were included for testing only. Out of these, 15 were selected for systematic sequence manipulations based on the amount of experimental evidence supporting them, lack of stop codons in the relevant reading frames, biological significance, and mechanistic diversity (i.e., covering +1, −1, and −2 frameshifting sites). The HIV and OAZ frameshifting sites were tested beforehand in isolation to confirm that our experimental setup is able to reproduce well established cases of −1, as well as +1 PRF.

**Synthetic library design—individual subsets**. For each of the subsets in the libraries, systematic sequence manipulations were performed on the set of 15

previously reported frameshifting sites. Design of subsets was carried out in Python.

*Multiple barcode controls*: We added multiple variants to the library that contained the same variable region, but different barcodes, in order to gauge potential effects of the barcode and the technical noise of our assay.

*Slippery site*: We tested all possible variations of the different slippery site patterns found in previously reported frameshifting events (XXXYYYZ, XYYZZZZ, influenza, and OAZ (for +1 PRF)). In the case of OAZ, both the first and the second codon in the slippery site UCC UGA were replaced with all other codons. In addition, we created all possible point mutations in −1 PRF slippery sites that would not give rise to a stop codon.

*Immediate downstream region/spacer*: We created all possible point mutations in the six nucleotides downstream of PRF slippery sites that would not give rise to a stop codon. In addition, we removed up to 6 bps immediately downstream (in 1 or 3 bp increments, depending on the presence of stop codons in downstream reading frames) or introduced up to 12 bps from five different constant sequences (in 1 or 3 bp increments, depending on the presence of stop codons in downstream reading frames).

*Immediate upstream region*: The first two codons preceding the slippery site (in the original frame) were replaced with all other codons (except stop codons).

*Upstream region*: The upstream region (excluding the last two codons before the slippery site) was randomly recoded five times. Moreover, variants were added for all 15 PRF events, in which the same part of the upstream region was replaced by four different constant sequences in increments of 6 bps, starting from the beginning of the variable region. In addition, we included the following structural variations in the design: we designed sequences 30 nt in length predicted (using ViennaRNA 2.0 (ref. [55]) and antaRNA[50,51]) to fold into the respective native structure (but having a different primary sequence or to have no secondary structure, a 9 bp long stem-loop at the 5′ or at the 3′ end (six sequences per set) and replaced the endogenous region in all 15 PRF events.

*Downstream region*: Variants were added for all 15 PRF events, in which the same part of the downstream region was replaced by three different constant sequences (different sets were used for predicted +1 and −1 PRF sites to avoid stop codons in the relevant frames) in increments of 6 bps, starting from the end of the variable region. For scanning mutagenesis, 30 positions (leaving the first 6 bp after the slippery site unchanged) were replaced with either A or C (G and T were avoided as they often give rise to stop codons in one of the frames). In addition, we included the following structural variations in the design: we used ViennaRNA 2.0 or pKiss[56] (which also predicts pseudoknots) to determine the native secondary structure of each of the 15 PRF events. We then created for each PRF event two sets of ten sequences each (using antaRNA) that are predicted to fold into the downstream secondary structure as determined using ViennaRNA and pKiss, respectively. Furthermore, we designed variants of the predicted HIV, SRV1, and nsp2F (PRRSV) downstream secondary structure, e.g., by extending or reducing the length of hairpins, and completely synthetic secondary structures and again used antaRNA to create sets of corresponding sequences that we used to replace the native downstream structure in all 15 PRF events.

*Completely synthetic variants*: We designed a set of four upstream regions, three potential slippery sites (resembling previously reported cases of the patterns XXXYYYZ, XYYZZZZ, and XXXYYYY), and created all possible combinations with a set of 57 artificial downstream sequences.

*Frameshifts*: We created variants, in which the slippery site was moved by one, two, or three positions in the 5′ or 3′ direction, in order to verify the frame dependence of the effect.

*Combinatorial variants*: Upstream, slippery site, and downstream regions from the whole set of 15 PRF sites were recombined, either preserving the original sequence fully or replacing potential stop codons in all frames with TGG.

**Synthetic library design—HIV gag-pol PRF sites**. We retrieved >30,000 full-length sequences of the HIV genome from clinical isolates (http://www.hiv.lanl.gov/) and selected gag-pol frameshifting sites that had the canonical sequence around the slippery site (TAATTTTTA), but differed in at least one position in the 30 nt upstream or the 120 nt downstream. We chose to exclude variability in the slippery site to screen for more subtle differences in frameshifting rates across clinical isolates.

**K562 cell culture**. K562 cells were acquired from ATCC. Cells were grown in Iscove's Modified Dulbecco Medium supplemented with 10% fetal bovine serum (SIGMA) and 1% penicillin–streptomycin solution (SIGMA). The cells were split when reaching a concentration of ~$10^6$ cells/ml. The cells were grown in an incubator at 37 °C and 5% $CO_2$. Cells were frozen in batches of $4 \times 10^6$ cells in growth medium supplemented with 5% DMSO.

**Construction of the master plasmid**. Master plasmids for library insertion were constructed by amplifying parts from existing vectors and cloning the parts sequentially into pZDonor 3.1. The master plasmid contained the EF1alpha promoter, *mCherry*, a designed multiple cloning site containing restriction sites for library cloning (RsrII and AscI), *gfp* and the SV40 terminator sequence. In the case of the master plasmid for cloning the stop-vector library, two complementary DNA

oligonucleotides (GACTGATAGCTGACTAGTCG and GTCCGACTAGTCAGC-TATCAG) were synthesized (IDT), annealed and introduced into the vector described above (cut with RsrII) such that the mCherry-proximal RsrII site was destroyed, and the downstream RsrII site (after the inserted stop cassette) was subsequently used for inserting the library.

**Synthetic library cloning**. The cloning steps were performed essentially as described previously[24,25]. We used Agilent oligo library synthesis technology to produce a pool of 17,809 different fully designed single-stranded 210-oligomers (Agilent Technologies, Santa Clara, CA), which was provided as a single pool of oligonucleotides (10 pmol). The two subsets of this pool corresponding to the designed and native libraries tested here were defined by unique amplification primers. The reverse primer contained a spacer of either 4 or 5 nucleotides to create the offset in the *gfp* reading frame. The oligo pool was dissolved in 200 µl Tris-ethylene-diamine-tetraacetic acid (Tris-EDTA) and then diluted 1:50 with Tris-EDTA, which was used as template for PCR. We amplified each of the four libraries by performing eight PCR reactions, each of which contained 19 µl of water, 5 µl of DNA, 10 µl of 5× Herculase II reaction buffer, 5 µl of 2.5 mM deoxynucleotide triphosphate each, 5 µl of 10 µM forward primer, 5 µl of 10 µM reverse primer, and 1 µl Herculase II fusion DNA polymerase (Agilent Technologies). The parameters for PCR were 95 °C for 1 min, 14 cycles of 95 °C for 20 s, and 68 °C for 1 min, each, and finally one cycle of 68 °C for 4 min. The oligonucleotides were amplified using library-specific common primers in the length of 35 nt, which have 18-nt complementary sequence to the single-stranded 210-mers and a tail of 17 nt containing RsrII (forward primer) and AscI (reverse primer) restriction sites. The PCR products were concentrated using Amicon Ultra, 0.5 ml 30 K centrifugal filters (Merck Millipore). The concentrated DNA was then purified using a PCR mini-elute purification kit (Qiagen) according to the manufacturer's protocol. Purified library DNA (540 ng total) was cut with the unique restriction enzymes RsrII and AscI (Fermentas FastDigest) for 2 h at 37 °C in two 40-µl reactions containing 4 µl fast digest (FD) buffer, 1 µl RsrII enzyme, 1 µl AscI enzyme, 18 µl DNA (15 ng/µl), and 16 µl water, followed by heat inactivation for 20 min at 65 °C. Digested DNA was separated from smaller fragments and uncut PCR products by electrophoresis on a 2.5% agarose gel stained with GelStar (Cambrex Bio Science Rockland). Fragments were cut from the gel and eluted using electroelution Midi GeBAflex tubes (GeBA, Kfar Hanagid, Israel). Eluted DNA was precipitated using sodium acetate–isopropanol. The master plasmid was cut with RsrII and AscI (Fermentas FastDigest) in a reaction mixture containing 6 µl FD buffer, 3 µl of each enzyme, and 3.5 µg of the plasmid in a total volume of 60 µl. After incubation for 2.5 h at 37 °C, 3 µl FD buffer, 3 µl alkaline phosphatase (Fermentas), and 24 µl water were added and the reactions were incubated for an additional 30 mins at 37 °C followed by 20 min at 65 °C. Digested DNA was purified using a PCR purification kit (Qiagen). The digested plasmid and DNA library were ligated for 30 min at room temperature in a 10 µl reactions, containing 150 ng plasmid and the library in a molar ratio of 1:1, 1 µl CloneDirect 10× ligation buffer, and 1 µl CloneSmart DNA ligase (Lucigen Corporation), followed by heat inactivation for 15 min at 70 °C. Ligated DNA was transformed into *E. coli* 10 G electrocompetent cells (Lucigen) divided into aliquots (23 µl each, plus 2 µl of the ligation mix), which were then plated on four Luria broth (LB) agar (200 mg/ml amp) 15-cm plates per trans-formation reaction (25 µl). The rationally designed (12,809 variants) and the native (5000 variants) parts of the library were cloned separately. For the rationally designed and the native library we collected ~$10^6$ and $4 \times 10^5$ colonies, respectively, for each frame the day after transformation by scraping the plates into LB medium. Library-pooled plasmids were purified using a NucleoBond Xtra maxi kit (Macherey Nagel). To ensure that the collected plasmids contain only a single insert of the right size, we performed colony PCR (at least 48 random colonies per library).

**Transfection into K562 cells and genomic integration**. The purified plasmid libraries were transfected into K562 cells and genomically integrated using the ZFN system for site-specific integration and the CompoZr® Targeted Integration Kit—AAVS1 (SIGMA). Transfections were carried out using Amaxa® Cell Line Nucleofector® Kit V (LONZA). To ensure library representation, we performed 15 nucleofections of the purified plasmid library for each rationally designed library (frame −1 and frame +1) and five for each native library. For each nucleofection, $4 \times 10^6$ cells were centrifuged and washed twice with 20 ml of Hank's Balanced Salt Solution (SIGMA). Cells were resuspended in 100 µl solution (warmed to room temperature) composed of 82 µl solution V and 19 µl supplement (Amaxa® Cell Line Nucleofector® Kit V). Next, the cells were mixed with 2.75 µg of donor plasmid and 0.6 µg ZFN mRNA (prepared in-house) just prior to transfection. Nucleofection was carried out using program T-16 on the NucleofectorTM device, immediately mixed with ~0.5 ml of precultured growth medium and transferred to a six well plate with additional 1.5 ml of precultured growth medium. A purified plasmid library was also transfected without the addition of ZFN and served as a control to determine when cells lost nonintegrated plasmids.

**Sorting the library by FACS**. K562 cells were grown for at least 14 days to ensure that nonintegrated plasmid DNA was eliminated. A day prior to sorting, cells were split to ~$0.25 \times 10^6$ cells/ml. On the day of sorting, cells were centrifuged,

resuspended in sterile PBS, and filtered using cell-strainer capped tubes (Becton Dickinson (BD) Falcon). Sorting was performed with BD FACSAria II SORP (special-order research product) at low sample flow rate and a sorting speed of ~18,000 cells/s. To sort cells that integrated the reporter construct successfully and in a single copy (~4% of the population), we determined a gate according to mCherry fluorescence so that only mCherry-expressing cells corresponding to a single copy of the construct were sorted (mCherry single population). We collected ~350 cells/variant on average for each library, in order to ensure adequate library representation.

Cells sorted for single integration of the transgene were grown for a week before we sorted the population (combined rationally designed and native library, but separately for −1 PRF reporters, +1 PRF reporters, and the stop-vector control) into 16 (in the case of the stop-vector 12) bins according to GFP fluorescence, after gating for a narrow range of mCherry expression to avoid effects coming from the influence of the variable region on overall expression level. This resulted in the loss of a significant fraction of variants (as only ~50% of the cells fell into the narrow range of mCherry fluorescence), but ensured that the subsequent sorting by GFP fluorescence was not biased by expression level. Each bin was defined to span a range of GFP fluorescence values such that it contains between 1 and 10% of the cell population. We collected ~1000 cells/variant on average in order to ensure adequate library representation. Cells from each bin were grown separately for freezing and purification of genomic DNA.

**Genomic DNA purification, amplification, and sample preparation**. For each of the bins, we purified genomic DNA by centrifuging $5 \times 10^6$ cells, washing them with 1 ml PBS and purifying DNA using DNeasy Blood and Tissue Kit (Qiagen) according to the manufacturer's protocol. In order to maintain the complexity of the library amplified from gDNA, PCR reactions were carried out on a gDNA amount calculated to contain a minimum average of 200 copies of each oligo included in the sample. For each of the 16 bins, we used 15 μg of gDNA as template in a two-step nested PCR. In the first step, three reactions were performed and each reaction contained 5 μg gDNA, 25 μl Kapa Hifi ready mix X2 (KAPA Biosystems), 2.5 μl 10 μM 5′ primer, and 2.5 μl of 10 μM 3′ primer. The parameters for the first PCR were 95 °C for 5 min, 18 cycles of 94 °C for 30 s, 65 °C for 30 s, and 72 °C for 60 s, each, and 1 cycle of 72 °C for 5 min. In the second PCR step, each reaction contained 2.5 μl of the first PCR product, 25 μl of Kapa Hifi ready mix X2 (KAPA Biosystems), 2.5 μl 10 μM 5′ primer, and 2.5 μl 10 μM 3′ primer. The PCR program was 95 °C for 5 min, 24 cycles of 94 °C for 30 s and 72 °C for 30 s, each, and 1 cycle of 72 °C for 5 min. Specific primers corresponding to the constant region of the plasmid were used. The 5′ primer contained a unique upstream 8-nt bin barcode sequence, and three different barcodes were used for each bin. The 3′ primer was common to all bins. Multiple PCR reaction products of each bin were combined. The concentration of the PCR samples was measured using a monochromator (Tecan i-control), and the samples were mixed in ratios corresponding to their ratio in the population, as defined when sorting the cells into the 16 bins. Sample preparation including gel elution and purification were performed as described above for amplicons from cDNA.

**RNA purification, cDNA synthesis, and sample preparation**. For the cell population sorted for single integration of the reporter construct, we performed RNA purification by centrifuging $10^7$ cells, washing them with PBS, splitting into two tubes, and purifying RNA using NucleoSpin RNA II kit (MACHEREY-NAGEL) according to the manufacturer's protocol. We prepared cDNA in two reverse transcription reaction for each replicate, using SuperScript® III First-Strand Synthesis System (Thermo Fisher Scientific) with random hexamer primers and 5 μg of input RNA (per reaction) according to the manufacturer protocol. For amplification of the library variants, three PCR reactions of 50 μl total volume were performed. Each reaction contained 5 μl cDNA, 25 μl of Kapa Hifi ready mix X2 (KAPA Biosystems), 2.5 μl 10 μM 5′ primer, and 2.5 μl 10 μM 3′ primer. The PCR program was 95 °C for 5 min, 20 cycles of 94 °C for 30 s and 72 °C for 30 s, each, and 1 cycle of 72 °C for 5 min. Specific primers corresponding to the constant region upstream and downstream of the splice sites were used. The PCR products were separated from potential unspecific fragments by electrophoresis on a 1.5% agarose gel stained with EtBr, cut from the gel, and cleaned in two steps: gel extraction kit (Qiagen) and SPRI beads (Agencourt AMPure XP). The sample was assessed for size and purity at the Tapestation, using high sensitivity D1K screenTape (Agilent Technologies, Santa Clara, California). We used 20 ng library DNA for library preparation for NGS; specific Illumina adaptors were added, and DNA was amplified using 14 amplification cycles. The sample was reanalyzed using Tapestation.

For RT-PCR analysis, we performed RNA purification on 5 million cells from individually constructed and integrated reporter constructs, followed by cDNA synthesis as described above (one reaction). PCR was performed using primers within the mCherry and GFP coding region, yielding a PCR product of ~1000 bp length. Forty cycles of PCR were used to obtain a strong signal even for less prominent bands. PCR products were analyzed on a 1% agarose gel.

**Mapping next generation sequencing reads**. To unambiguously identify the variant of origin, a unique 12-mer barcode sequence was placed at the 5′ end of

each variable region. DNA was sequenced on a NextSeq-500 sequencer. We determined for each read (PE150) its bin barcode and its variant barcode, and discarded all the reads that could not be assigned to a bin and a library variant of origin. In a second step, for each variant the mapped reads (paired-end, 150 bases for each mate) were aligned to the designed sequence (in the context of the reporter) and only those ones were counted that mapped without a mismatch over the whole length (disregarding the last five bases). Reads showing a mismatch, in particular a single nucleotide deletion or insertion, were quantified separately and if they passed the same threshold applied to correct library variants a PRF readout (% GFP fluorescence) was calculated for them. These additional variants were only used in comparisons with the original designed variant and not incorporated in the general pool of library variants considered for the analyses. For all three libraries (−1 PRF reporters, +1 PRF reporters, and the stop-vector control), ~40 milllion successfully mapped reads were used for subsequent analysis. In addition, we sequenced the entire cell population using greater read lengths (PE300) to cover the entire length of the variable region and to obtain a measure of the fraction of reads per variant containing a synthesis or cloning error. Mapping was performed using custom-made Python scripts.

**Computing a measure of frameshifting efficiency**. We applied a number of filters to the raw sequencing data to reduce experimental noise. First, variants with <20 reads (perfect matches along the whole length) mapped across bins were removed. Second, for bins with a read count of less than three, the bin value was set to zero. Third, for each variant we set to zero bins surrounded by zero values as these constituted isolated bins unlikely to come from the actual distribution. Forth, to reduce bias coming from the open bins at the extreme values, we set their count to match their neighbor's if it was higher, as these bins are defined as containing the tails of the distribution of variants with no GFP fluorescence and maximal GFP expression, respectively, and peaks in these extreme bins are considered experimental noise. For each variant, we normalized the values across the 16 bins by dividing the filtered read counts for each bin by the sum of all filtered read counts. To determine whether a distribution is unimodal, we applied a Savitzky–Golay filter for smoothing the data and removing local maxima introduced by experimental noise. We detected peaks in the smoothed distribution by a simple approach, in which a point is considered a maximum peak if it has the maximal value, and was preceded by a value lower by 5%. Variants without detectable peaks (due to the read number not reaching the threshold) were excluded at this stage. Variants with more than one peak were filtered out at the level of analysis (but are contained in Supplementary Data 2). For each bin, we calculated the median of the log2 of GFP fluorescence intensity as measured by FACS for all the cells sorted into that bin, and used this as the GFP fluorescence value associated with this bin. For each variant with one peak after smoothing, we calculated the weighted average for the distribution of reads across bins (using unsmoothed read counts normalized for each variant and the GFP fluorescence value for the bins, as described above), resulting in what is referred to in the main text and figures as the GFP fluorescence (in log2).

As described in the main text and illustrated in Supplementary Fig. 2a–c, we set the lowest value obtained (the variant with the lowest weighted average of the distribution) to 0% GFP fluorescence, and the highest value obtained (the variant with highest weighted average of the distribution) to 100% GFP fluorescence. Accordingly, a percentage was assigned to each variant based on the GFP fluorescence (log2 of the weighted average of the reads distribution) of a variant as follows:

$$[\% \text{ GFP fluorescence}]_{\text{variant X}} = ([\text{GFP fluorescence}]_{\text{variant X}} - \min([\text{GFP fluorescence}]_{\text{all variants}}))/(\max([\text{GFP fluorescence}]_{\text{all variants}}) - \min([\text{GFP fluorescence}]_{\text{all variants}})) \times 100.$$

% wild-type GFP fluorescence of a variant was calculated as follows:

$$[\% \text{ wild-type GFP fluorescence}]_{\text{variant X}} = [\% \text{ GFP fluorescence}]_{\text{variant X}}/[\% \text{ GFP fluorescence}]_{\text{wild-type version of variant X}} \times 100.$$

As a measure for steady-state RNA expression levels, we computed the log ratio of RNA/DNA reads for all variants with at least 20 DNA reads past filtering: $\log2([\text{RNA reads}]_{\text{variant X}}/[\text{DNA reads}]_{\text{variant X}})$.

**Machine learning approaches**. Machine learning procedures were carried out using the python scikit-learn (version 0.18.2) and XGBoost package. Initially, from all duplicated sequences (e.g., barcode control sets), which passed filtering, a single variant was randomly chosen for all subsequent steps to avoid biases resulting from having duplicated sequences. We created sets of variants corresponding either to the full rationally designed library (full library), the −1 PRF sites from the main set of events tested positive in our assay combined (frameshifting events) or to all variants based on one of these PRF events in isolation. A total of 10% of the variants in the individual sets (20% in case of the HIV variants) were put aside and used only for evaluation of models built using the other 90% (80%). We chose Gradient Boosting Decision Trees (XGBoost[63]) as the prediction algorithm because it can capture nonlinear interactions between features, allows combining different types of continuous and categorical features, and has proven to be a powerful approach in predicting the effect of regulatory regions[24,25].

For predictions based on secondary structure, we used the fold function from the Vienna RNA package 2.0 and extracted both the minimal free energy ("dg"; calculated for different lengths of downstream (10–120 nt) regions) and the

predicted pairedness for each position ("sec"; for each position in the downstream region this set of features contains the information whether the position is predicted (Vienna RNAfold) to be paired or not (0 or 1)).

For predictions based on tAI, we computed (based on Waldman et al.[91]) the tAI of codons in the original and −1 frame, as well as the (per position) difference between frame −1 and frame 0, and used the information for 14 starting from the first codon from the slippery site (in the case of the canonical −1 pattern XXXYYYZ, XXY in frame 0 and XXX in frame −1) as features.

For predictions based on amino acid identity, information on the type of amino acids around the slippery site (from position −3 relative to the site of frameshifting until the end of the slippery site) was used (unpolar, polar, or charged).

Different hyperparameter settings for learning rate, n_estimators and max_depth were tested in a systematic and combinatorial fashion, using tenfold cross-validation for each set of variants and features independently. Typically, ~100 tests were performed and the best set of hyperparameters used for subsequent steps.

At the end, the model was tested by training it on the entire training set (90% of all relevant unique library variants, 80% in the case of HIV clinical isolates) and scoring the correlation (Pearson) of predicted vs. measured values based on the held-out test set (10% of relevant unique library variants from the relevant set, 20% in the case of HIV clinical isolates), which had not been used at any stage during development of the model. This approach was used for each of the features sets in isolation, as well as all of the features combined.

For classification, the same pipeline was used. Here, XGBClassifier was trained and the performance scored on the test set, reporting the area under the ROC curve as a metric.

Feature importance and effect on the model was determined using SHAP analysis[64].

**General data analysis**. For data analysis, we used python 2.7.11 with pandas 0.20.3, numpy 1.13.1, seaborn 0.6, scipy 0.17, scikit-learn 0.18.2, and shap 0.28.5. Confidence intervals were calculated by bootstrapping (1000 iterations).

**Reporting summary**. Further information on research design is available in the Nature Research Reporting Summary linked to this article.

## Data availability
All data generated in this study are available in the NCBI gene expression omnibus (GEO) under accession GSE145684. Information on PRF sites in viral, bacterial, and eukaryotic genomes was gathered from FSDB[28] (http://wilab.inha.ac.kr/fsdb/) and the cited literature. HIV sequences have been retrieved from the HIV sequence database (http://www.hiv.lanl.gov/). All data are available from the corresponding author upon reasonable request.

## Code availability
The code used to process the data, generate the figures, and train and test the predictive models are available as a GitHub repository (https://github.com/martinmikl/PRF_mpra).

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

## Acknowledgements
The authors thank Adina Weinberger, Orna Dahan, Alexey Gritsenko, and Roni Rak for helpful discussions, Ronit Nir, Tali Avnit-Sagi, and Maya Lotan-Pompan for technical advice, and the Moshe Oren lab for advice and reagents. This work was supported by an EMBO long-term fellowship (to M.M.). E.S. is supported by the Crown Human Genome Center; the Else Kroener Fresenius Foundation; D.L. Schwarz; J.N. Halpern; L. Steinberg; J. Benattar; Aliza Moussaieff; Adelis Foundation; and grants funded by the European Research Council and the Israel Science Foundation.

## Author contributions
Conceptualization: M.M., Y.P., and E.S.; methodology, software, and formal analysis: M.M.; investigation: M.M.; writing: M.M., Y.P., and E.S.; funding acquisition: M.M., Y.P., and E.S.; and supervision: Y.P. and E.S.

## Competing interests
The authors declare no competing interests.
