## [Peer Review File · Nature Communications]

REVIEWERS' COMMENTS:

Reviewer #1 (Remarks to the Author):

In this manuscript, Mikl et al describe a new fluorescent-based reporter for high-throughput assessment of sequence variations introduced into frameshifting cassettes (frameshift site and stimulatory elements). They have also demonstrated the applicability of this system for testing the sequence constraints of many frameshifting cassettes operational in human cells (of viral and cellular origins). The manuscript has been transferred from another journal and I reviewed two previous versions of the manuscript. The current version is substantially different from the previous version as the authors dropped the claim regarding the discovery of novel cases of ribosomal frameshifting and related parts of this work. I respect and support this decision. This claim requires more substantial experimental validation than what was presented in the previous version and if the authors indeed will pursue this endeavor, I sincerely hope that my previous review will be of help. As for the current manuscript, I think the method described here will be very useful for studying ribosomal frameshifting (as well as of other recoding mechanisms) and indeed could be used as a “fishing” tool for identification of a pool of potential low-confidence candidate frameshifting events. My further suggestions for improving the current manuscript are comparatively minor.

Page 2. Lanes 45-46: “Most human cases known to date were found serendipitously or through homologous genes.”

This statement is a bit problematic. Indeed, in general, some cases of frameshifting were discovered serendipitously, but not most. Most were predicted based on sequence analysis and then confirmed experimentally, this includes even the first cellular example, bacterial release factor 2. First, it was predicted (Craigen et al 1985) and only then confirmed (Craigen & Caskey 1986). If we limit ourselves to human genes only, then we have only a handful of examples, oaz1 frameshifting was first characterized in rat and then found in humans, oaz2 and oaz3 were found as paralogs of oaz1. Frameshifting in PNMA3 was discovered as a result of a systematic bioinformatics search. Even proclaimed frameshifting in CCR5 stemmed from an attempt to identify novel cases of ribosomal frameshifting systematically. Perhaps only Edr/PEG10 might classify as serendipitous discovery and even that is a bit of an overstretch because frameshifting was already suspected because of the two long overlapping ORFs in the reconstructed mRNA sequence. Besides, strictly speaking it was first characterized in mice. Outside of humans, most of the recently discovered frameshifting cases (or other recoding events) follow the same trajectory: phylogenetic analysis followed by experimental validation, see publications by Andrew Firth and Manolis Kellis labs. To me one potential advantage of Mikl et al approach is that it may potentially enable discovery of frameshifting cases whose identification is not possible using phylogenetic analysis, e.g. recently evolved cases.

In relation to the description of the human cases the authors correctly cite the discovery of PNMA3, oaz1, and oaz2. Oaz3 was discovered by two labs independently, but the authors omitted Ivanov et al (2000). The reference on frameshifting discovery in Edr/PEG10 (Shigemoto et al 2001) is also missing.

Some of the observations made by the authors (including frameshifting efficiencies) differ from what was published earlier. This is not surprising because experimental methods have their own unique limitations and considerable discrepancies in measured frameshifting efficiencies have been often observed in the past. The authors' description of the method limitations is far better now than what was in the original manuscript, however, many factors that may be responsible for inaccurate frameshifting measurements are not mentioned. I am sure that we don't even know many such factors, but perhaps the authors will find some of the following useful and worthy mentioning in the manuscript while discussing the limitations of their approach.

1. The effect of protein sequence encoded in the tested cassette on reporter activity or its stability, see Loughran et al (2017).
2. Protein factors involved in modulation of frameshifting, see Naphthine et al (2019). Such proteins may not be present in the cell line where frameshifting is being tested.
3. Concentrations of specific metabolites, e.g. the ribosomal frameshifting in testis-specific antizyme 3 is very low in other cells, see Howard et al (2001). This is probably because polyamine levels in germ cells are far higher where endogenous levels of oaz3 would also be expected to be higher.
4. Distance between the ribosomes, see Smith et al (2019).
5. Expression levels, see Gurvich et al (2005)

While the last two examples relate to frameshifting in bacteria, there is no reason to believe that such factors would be irrelevant in human cells. Speaking of yet unknown factors we could speculate that co- and posttranscriptional RNA modifications may affect frameshifting but may not be reproduced in the reporter constructs, e.g. it has been shown that inosines cause ribosome pauses, see (Licht et al 2019)

In relation to CCR5 (Lines 190-194), I think it could be helpful to mention that the reported frameshifting in CCR5 is simply an artifact of a dual-luciferase reporter as has been shown recently by Khan et al (2019). I suggest that the authors should decide on whether to cite Khan et al (2019) in consultation with the editor whom I will provide with additional confidential information that I cannot mention here .

References:

Craig WJ, Caskey CT. Expression of peptide chain release factor 2 requires high-efficiency frameshift. *Nature*. 1986 Jul 17-23;322(6076):273-5. PMID: 3736654

Craig WJ, Cook RG, Tate WP, Caskey CT. Bacterial peptide chain release factors: conserved primary structure and possible frameshift regulation of release factor 2. *Proc Natl Acad Sci U S A*. 1985 Jun;82(11):3616-20. PMID: 3889910

Gurvich OL, Baranov PV, Gesteland RF, Atkins JF. Expression levels influence ribosomal frameshifting at the tandem rare arginine codons AGG_AGG and AGA_AGA in *Escherichia coli*. *J Bacteriol*. 2005 Jun;187(12):4023-32. PMID: 15937165

Howard MT, Shirts BH, Zhou J, Carlson CL, Matsufuji S, Gesteland RF, Weeks RS, Atkins JF. Cell culture analysis of the regulatory frameshift event required for the expression of mammalian antizymes. *Genes Cells*. 2001 Nov;6(11):931-41. PMID: 11733031.

Ivanov IP, Rohrwasser A, Terreros DA, Gesteland RF, Atkins JF. Discovery of a spermatogenesis stage-specific ornithine decarboxylase antizyme: antizyme 3. *Proc Natl Acad Sci U S A*. 2000 Apr 25;97(9):4808-13. PMID: 10781085

Khan YA, Loughran G, Atkins JF. Contesting the evidence for -1 frameshifting in immune-functioning C-C chemokine receptor 5 (CCR5) – the HIV-1 co-receptor. <https://doi.org/10.1101/513333>

Licht K, Hartl M, Amman F, Anrather D, Janisiw MP, Jantsch MF. Inosine induces context-dependent recoding and translational stalling. *Nucleic Acids Res*. 2019 Jan 10;47(1):3-14. doi: 10.1093/nar/gky1163. PMID: 30462291

Loughran G, Howard MT, Firth AE, Atkins JF. Avoidance of reporter assay distortions from fused dual reporters. *RNA*. 2017;23(8):1285–1289. doi:10.1261/rna.061051.117

Napthine S, Bell S, Hill CH, Brierley I, Firth AE. Characterization of the stimulators of protein-directed ribosomal frameshifting in Theiler's murine encephalomyelitis virus. *Nucleic Acids Res*. 2019 Sep 5;47(15):8207-8223. doi: 10.1093/nar/gkz503.

Shigemoto K, Brennan J, Walls E, Watson CJ, Stott D, Rigby PW, Reith AD. Identification and characterisation of a developmentally regulated mammalian gene that utilises -1 programmed ribosomal frameshifting. *Nucleic Acids Res*. 2001 Oct 1;29(19):4079-88. PMID: 11574691

Smith AM, Costello MS, Kettring AH, Wingo RJ, Moore SD. Ribosome collisions alter frameshifting at translational reprogramming motifs in bacterial mRNAs. *Proc Natl Acad Sci U S A*. 2019 Oct 22;116(43):21769-21779. doi: 10.1073/pnas.1910613116. Epub 2019 Oct 7.

//Pasha Baranov//

Reviewer #2 (Remarks to the Author):

This revised version of the manuscript is wonderful. Although not perfect, mostly because of the sheer number of constructs tested and volume of data, I do believe that this will represent an important addition to the field. The authors are to be commended for their work.

POINT-BY-POINT RESPONSE TO THE REVIEWERS' COMMENTS:

Reviewer #1 (Remarks to the Author):

In this manuscript, Mikl et al describe a new fluorescent-based reporter for high-throughput assessment of sequence variations introduced into frameshifting cassettes (frameshift site and stimulatory elements). They have also demonstrated the applicability of this system for testing the sequence constraints of many frameshifting cassettes operational in human cells (of viral and cellular origins). The manuscript has been transferred from another journal and I reviewed two previous versions of the manuscript. The current version is substantially different from the previous version as the authors dropped the claim regarding the discovery of novel cases of ribosomal frameshifting and related parts of this work. I respect and support this decision. This claim requires more substantial experimental validation than what was presented in the previous version and if the authors indeed will pursue this endeavor, I sincerely hope that my previous review will be of help. As for the current manuscript, I think the method described here will be very useful for studying ribosomal frameshifting (as well as of other recoding mechanisms) and indeed could be used as a “fishing” tool for identification of a pool of potential low-confidence candidate frameshifting events. My further suggestions for improving the current manuscript are comparatively minor.

Page 2. Lanes 45-46: “Most human cases known to date were found serendipitously or through homologous genes.”

This statement is a bit problematic. Indeed, in general, some cases of frameshifting were discovered serendipitously, but not most. Most were predicted based on sequence analysis and then confirmed experimentally, this includes even the first cellular example, bacterial release factor 2. First, it was predicted (Craigén et al 1985) and only then confirmed (Craigén & Caskey 1986). If we limit ourselves to human genes only, then we have only a handful of examples, oaz1 frameshifting was first characterized in rat and then found in humans, oaz2 and oaz3 were found as paralogs of oaz1. Frameshifting in PNMA3 was discovered as a result of a systematic bioinformatics search. Even proclaimed frameshifting in CCR5 stemmed from an attempt to identify novel cases of ribosomal frameshifting systematically. Perhaps only Edr/PEG10 might classify as serendipitous discovery and even that is a bit of an overstretch because frameshifting was already suspected because of the two long overlapping ORFs in the reconstructed mRNA sequence. Besides, strictly speaking it was first characterized in mice. Outside of humans, most of the recently discovered frameshifting cases (or other recoding events) follow the same trajectory: phylogenetic analysis followed by experimental validation, see publications by Andrew Firth and Manolis Kellis labs. To me one potential advantage of Mikl et al approach is that it may potentially enable discovery of frameshifting cases whose identification is not possible using phylogenetic analysis, e.g. recently evolved cases.

We changed this sentence according to the reviewer's suggestions to “Many of the human PRF events were found through homology.” (lines 41-42)

In relation to the description of the human cases the authors correctly cite the discovery of PNMA3, oaz1, and oaz2. Oaz3 was discovered by two labs independently, but the authors omitted Ivanov et al (2000). The reference on frameshifting discovery in Edr/PEG10 (Shigemoto et al 2001) is also missing.

We apologize for this omission and added the missing references in the revised manuscript (line 37).

Some of the observations made by the authors (including frameshifting efficiencies) differ from what was published earlier. This is not surprising because experimental methods have their own unique limitations and considerable discrepancies in measured frameshifting efficiencies have been often observed in the past. The authors' description of the method limitations is far better now than what was in the original manuscript, however, many factors that may be responsible for inaccurate frameshifting measurements are not mentioned. I am sure that we don't even know many such factors, but perhaps the authors will find some of the following useful and worthy mentioning in the manuscript while discussing the limitations of their approach.

1. The effect of protein sequence encoded in the tested cassette on reporter activity or its stability, see Loughran et al (2017).
2. Protein factors involved in modulation of frameshifting, see Naphine et al (2019). Such proteins may not be present in the cell line where frameshifting is being tested.
3. Concentrations of specific metabolites, e.g. the ribosomal frameshifting in testis-specific antizyme 3 is very low in other cells, see Howard et al (2001). This is probably because polyamine levels in germ cells are far higher where endogenous levels of oaz3 would also be expected to be higher.
4. Distance between the ribosomes, see Smith et al (2019).
5. Expression levels, see Gurvich et al (2005)

While the last two examples relate to frameshifting in bacteria, there is no reason to believe that such factors would be irrelevant in human cells. Speaking of yet unknown factors we could speculate that co- and posttranscriptional RNA modifications may affect frameshifting but may not be reproduced in the reporter constructs, e.g. it has been shown that inosines cause ribosome pauses, see (Licht et al 2019)

We thank the reviewer for their comments and mention these points in the revised manuscript "Moreover, many of the general caveats in using reporter systems apply also here, such as the non-native sequence context and expression levels, the effect of the tested sequence on

fluorescent readout and protein stability and the concentration of potential trans-acting proteins or metabolites in the particular cell type used in the experiment.” (lines 521-525)

In relation to CCR5 (Lines 190-194), I think it could be helpful to mention that the reported frameshifting in CCR5 is simply an artifact of a dual-luciferase reporter as has been shown recently by Khan et al (2019). I suggest that the authors should decide on whether to cite Khan et al (2019) in consultation with the editor whom I will provide with additional confidential information that I cannot mention here.

We mention in the revised manuscript that CCR5 frameshifting has been contested by Khan et al. (2019).

References:

Craig WJ, Caskey CT. Expression of peptide chain release factor 2 requires high-efficiency frameshift. *Nature*. 1986 Jul 17-23;322(6076):273-5. PMID: 3736654

Craig WJ, Cook RG, Tate WP, Caskey CT. Bacterial peptide chain release factors: conserved primary structure and possible frameshift regulation of release factor 2. *Proc Natl Acad Sci U S A*. 1985 Jun;82(11):3616-20. PMID: 3889910

Gurvich OL, Baranov PV, Gesteland RF, Atkins JF. Expression levels influence ribosomal frameshifting at the tandem rare arginine codons AGG_AGG and AGA_AGA in *Escherichia coli*. *J Bacteriol*. 2005 Jun;187(12):4023-32. PMID: 15937165

Howard MT, Shirts BH, Zhou J, Carlson CL, Matsufuji S, Gesteland RF, Weeks RS, Atkins JF. Cell culture analysis of the regulatory frameshift event required for the expression of mammalian antizymes. *Genes Cells*. 2001 Nov;6(11):931-41. PMID: 11733031.

Ivanov IP, Rohrwasser A, Terreros DA, Gesteland RF, Atkins JF. Discovery of a spermatogenesis stage-specific ornithine decarboxylase antizyme: antizyme 3. *Proc Natl Acad Sci U S A*. 2000 Apr 25;97(9):4808-13. PMID: 10781085

Khan YA, Loughran G, Atkins JF. Contesting the evidence for -1 frameshifting in immune-functioning C-C chemokine receptor 5 (CCR5) – the HIV-1 co-receptor. <https://doi.org/10.1101/513333>

Licht K, Hartl M, Amman F, Anrather D, Janisiw MP, Jantsch MF. Inosine induces context-dependent recoding and translational stalling. *Nucleic Acids Res*. 2019 Jan 10;47(1):3-14. doi: 10.1093/nar/gky1163. PMID: 30462291

Loughran G, Howard MT, Firth AE, Atkins JF. Avoidance of reporter assay distortions from fused dual reporters. *RNA*. 2017;23(8):1285–1289. doi:10.1261/rna.061051.117

Napthine S, Bell S, Hill CH, Brierley I, Firth AE. Characterization of the stimulators of protein-directed ribosomal frameshifting in Theiler's murine encephalomyelitis virus. *Nucleic Acids Res.* 2019 Sep 5;47(15):8207-8223. doi: 10.1093/nar/gkz503.

Shigemoto K, Brennan J, Walls E, Watson CJ, Stott D, Rigby PW, Reith AD. Identification and characterisation of a developmentally regulated mammalian gene that utilises -1 programmed ribosomal frameshifting. *Nucleic Acids Res.* 2001 Oct 1;29(19):4079-88. PMID: 11574691

Smith AM, Costello MS, Kettring AH, Wingo RJ, Moore SD. Ribosome collisions alter frameshifting at translational reprogramming motifs in bacterial mRNAs. *Proc Natl Acad Sci U S A.* 2019 Oct 22;116(43):21769-21779. doi: 10.1073/pnas.1910613116. Epub 2019 Oct 7.

//Pasha Baranov//

Reviewer #2 (Remarks to the Author):

This revised version of the manuscript is wonderful. Although not perfect, mostly because of the sheer number of constructs tested and volume of data, I do believe that this will represent an important addition to the field. The authors are to be commended for their work.

We thank the reviewer for his kind words and acknowledge the imperfections that come with a large-scale assay.